# DocVAL: Validated Chain-of-Thought Distillation for Grounded Document VQA

**Pinaki Prasad Guha Neogi** [* 1]  **Ahmad Mohammadshirazi** [* 1]  **Ser-Nam Lim** [2]  **Rajiv Ramnath** [1]

## Abstract

Document visual question answering requires models not only to answer questions correctly, but also to precisely localize answers within complex document layouts. While large vision-language models (VLMs) achieve strong spatial grounding, their inference cost and latency limit real-world deployment; on the other hand, compact VLMs are efficient but suffer substantial localization degradation under standard fine-tuning or distillation. To address this gap, we propose **DocVAL**, a validated chain-of-thought (CoT) distillation framework that transfers explicit spatial reasoning from large teacher models to compact, deployable student VLMs. DocVAL combines **(1)** teacher-generated spatial CoT supervision, **(2)** a rule-based dual-mode validator that filters low-quality training signals and provides fine-grained, pixel-level corrective feedback, and **(3)** a validation-driven two-stage training procedure with iterative refinement. Text detection is used only as training-time scaffolding for supervision and validation, enabling the final student to operate as a pure VLM without OCR or detection at inference. Across multiple document understanding benchmarks, the proposed **DocVAL** yields consistent improvements of up to **6-7 ANLS** points over comparable compact VLMs. We further introduce mean Average Precision (mAP) as a localization metric for document question answering and report strong spatial grounding performance under this new evaluation. We release **95K validator-verified CoT traces** and show that high-quality, validated supervision is more effective than scaling unfiltered data, enabling efficient and trustworthy document grounding. Code/Data: [GitHub](GitHub).

---

[*]Equal contribution  [1]Department of Computer Science and Engineering, Ohio State University, Ohio, US  [2]Department of Computer Science, University of Central Florida, Florida, US. Correspondence to: Pinaki Prasad Guha Neogi <guha-neogi.2@osu.edu>.

*Proceedings of the $43^{rd}$ International Conference on Machine Learning*, Seoul, South Korea. PMLR 306, 2026. Copyright 2026 by the author(s).

## 1. Introduction

Answering questions over documents such as invoices, receipts, forms, and scanned records requires models to reason jointly over textual content and spatial layout. Beyond predicting correct answers, practical document understanding systems must also precisely localize answers within the document, enabling trust, interpretability, and human verification in high-stakes domains such as finance, healthcare, and legal processing (Mathew et al., 2021; Park et al., 2019; Jaume et al., 2019).

Recent vision-language models (VLMs) have demonstrated strong performance on document understanding benchmarks that evaluate answer accuracy and layout reasoning (Huang et al., 2022; Luo et al., 2024; Liao et al., 2024). However, this progress exposes a fundamental deployment dilemma. Large VLMs achieve robust answer accuracy and spatial grounding, but incur high inference latency, large memory footprints, and reliance on cloud APIs, making them impractical for large-scale or on-device deployment. In contrast, compact VLMs are efficient and deployable, yet suffer severe degradation in spatial reasoning, particularly when precise bounding box localization is required.

This gap persists because current training paradigms are misaligned with the requirements of spatial grounding. Standard fine-tuning and knowledge distillation approaches transfer output distributions or intermediate representations (Ahn et al., 2019; Gou et al., 2021), but do not teach *how* localization decisions are made. As a result, compact models often produce plausible answers without reliably grounding them in the document. This issue is further compounded by commonly used evaluation metrics such as ANLS (Li & Liu, 2007), which emphasize textual correctness while largely ignoring spatial precision. The outcome is a growing *trust gap*: models appear accurate, yet fail to provide verifiable, location-aware predictions, which is particularly problematic in safety-critical domains such as healthcare, law, and finance, where hallucinated or weakly grounded outputs limit the reliability and practical adoption of such systems.

In this work, we argue that closing this gap requires transferring not just answers or coordinates, but *explicit spatial reasoning processes*. We introduce **DocVAL**, a validated chain-of-thought (CoT) distillation framework that enables

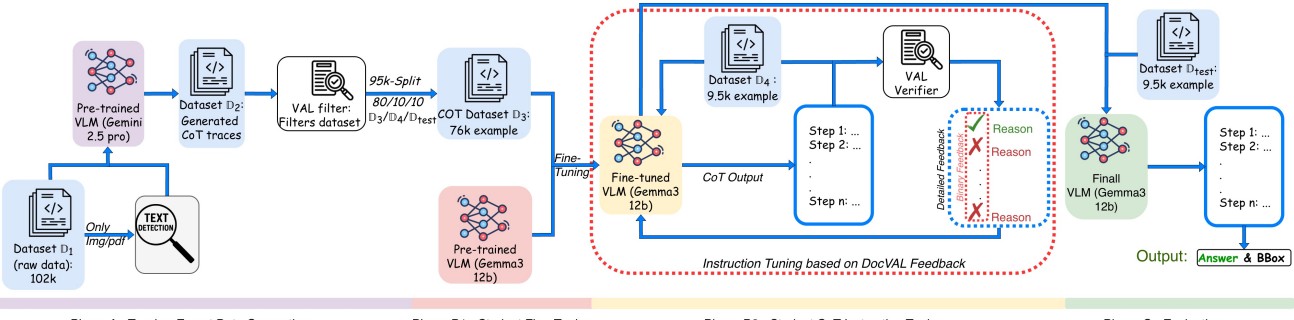

*Figure 1.* **Overview of the DocVAL Framework.** A three-phase pipeline for validated chain-of-thought (CoT) distillation. **Phase A: Teacher Data Generation** —The teacher VLM (Gemini 2.5 Pro) generates CoT traces from raw documents, which are filtered by DocVAL for quality assurance. **Phase B1: Student Fine-Tuning** —A smaller VLM (Gemma3-12B) is fine-tuned on the validated CoT dataset. **Phase B2: Instruction Tuning** —The student model iteratively refines its reasoning with DocVAL feedback until convergence. **Phase C: Evaluation** —The final model generates stepwise reasoning to produce the answer and bounding boxes.

compact VLMs to acquire strong document grounding capabilities by *explicitly supervising intermediate spatial reasoning steps* through quality-controlled reasoning transfer. Unlike generic visual reasoning, document spatial grounding is inherently compositional: models must identify relevant text regions, reason about their relative layout, and translate this reasoning into precise pixel-level coordinates. Chain-of-thought reasoning is particularly well-suited to this setting because it externalizes intermediate spatial decisions—such as locating candidate regions, verifying semantic alignment, and refining boundaries—that are otherwise implicit in end-to-end coordinate regression. Prior work has shown that CoT improves structured reasoning by exposing intermediate steps (Wei et al., 2022; Shao et al., 2025); in the document domain, these steps naturally correspond to spatial operations over layout and geometry. By supervising these intermediate spatial descriptions, rather than only final answers or bounding boxes, DocVAL provides a principled mechanism for transferring spatial reasoning processes from large teacher models to compact, deployable students. DocVAL further draws inspiration from validation-driven learning (Madaan et al., 2023), adapting it to the unique challenges of spatial reasoning in document understanding.

**Key insight.** Spatial grounding can be effectively distilled when four conditions are met: (1) teacher models express reasoning as explicit spatial descriptions rather than opaque coordinate predictions; (2) training data is rigorously filtered to remove hallucinated boxes and inconsistent reasoning; (3) student errors are corrected through actionable, pixel-level feedback rather than binary scores; and (4) students are trained to predict bounding boxes directly from visual inputs, without relying on detection or OCR at inference time. DocVAL is explicitly designed to satisfy all four conditions.

Figure 1 illustrates the DocVAL pipeline. In Phase A, a large teacher VLM generates spatial CoT traces that are validated using text detection solely as training-time scaffolding for supervision and filtering. In Phase B, a compact student is trained in two stages: first through supervised learning on validated CoT traces, and then through validation-driven iterative refinement using structured feedback. In Phase C, the final student performs document question answering with spatial grounding as a pure VLM, requiring no auxiliary models at inference.

Beyond the training framework, DocVAL introduces a new dataset of **95K high-quality, validator-verified CoT traces**, each aligned with answer text and bounding box annotations. To our knowledge, this is the first large-scale dataset that couples document-level chain-of-thought reasoning with explicit spatial supervision. Importantly, our experiments across multiple benchmark datasets demonstrate that *data quality outweighs data quantity*: training on larger but unvalidated reasoning data degrades localization performance, while a smaller set of validated examples yields substantially higher accuracy and efficiency. This finding highlights the importance of validation-aware dataset construction for reasoning-based fine-tuning.

Our main contributions are:

- We propose **DocVAL**, a validated chain-of-thought distillation framework that transfers spatial reasoning from large teacher models to compact, deployable VLMs.

- We incorporate **VAL**, a rule-based dual-mode validator, to adapt validation-driven learning to document spatial grounding by filtering low-quality supervision and providing fine-grained, pixel-level feedback for iterative refinement.

- We present a new dataset of **95K validator-verified CoT traces** for document question answering with spatial grounding.

- Through extensive ablations across multiple benchmarks and data scales, we show that *validation-driven, high-quality supervision is substantially more data-efficient than unvalidated training*, enabling strong spatial localization with significantly fewer training examples and without OCR or detection at inference.

The remainder of this paper is organized as follows: Section 2 reviews related work; Section 3 details the DocVAL framework; Section 4 describes experimental setup and datasets; Section 5 presents results and ablations; and Section 6 presents the conclusion.

## 2. Related Work

### 2.1. Document Understanding and Spatial Grounding

Early document understanding models such as LayoutLM (Xu et al., 2020a) and its successors (Xu et al., 2020b; Huang et al., 2022) introduced layout-aware representation learning through joint text, image, and layout pre-training. More recent approaches follow either OCR-free or OCR-based paradigms. OCR-free models (Kim et al., 2022; Lee et al., 2023; Mohammadshirazi et al., 2024) integrate text recognition end-to-end but often require high-resolution inputs and incur substantial computational cost. OCR-based methods achieve richer spatial integration by encoding bounding boxes or layout tokens (Lu et al., 2024; Liao et al., 2024; Luo et al., 2024), but typically rely on external OCR or detection pipelines at inference time. Recent agentic and graph-augmented document VQA systems further improve interpretability by decomposing document reasoning into specialized modules or explicitly modeling spatial relations, but they typically retain additional inference-time components for tool orchestration, graph reasoning, or memory access (Mohammadshirazi et al., 2025a;b).

Despite strong answer accuracy, most existing systems provide limited support for explicit, interpretable spatial grounding. Localization decisions are often implicit or tightly coupled to preprocessing modules, making verification difficult and introducing brittle inference dependencies. In contrast, our paper demonstrates that compact vision-language models can learn end-to-end document grounding when trained with appropriate reasoning supervision, without relying on detection at inference. We provide a detailed conceptual comparison of DocVAL relative to grounding-focused work such as DOGR (Zhou et al., 2025) and TRIG (Li et al., 2025) in Appendix F.1.

### 2.2. Reasoning, Distillation, and Validation

Chain-of-thought (CoT) prompting (Wei et al., 2022) and its extensions to vision-language models (Xu et al., 2025; Zheng et al., 2023) demonstrate that explicit step-by-step reasoning can improve model performance. However, existing approaches primarily focus on semantic or logical reasoning and provide limited support for document-specific spatial reasoning. Moreover, CoT traces are typically treated as supervision without verification, which is particularly problematic in document understanding, where unvalidated reasoning can include hallucinated regions, inconsistent spatial relations, or incorrect coordinate references that propagate noise during training.

This limitation is compounded in knowledge distillation settings. Traditional distillation methods (Gou et al., 2021; Ji et al., 2021; Lei & Tao, 2023) and recent VLM distillation approaches (Li et al., 2024; 2023) focus on transferring representations or output distributions, but do not explicitly teach how spatial localization decisions are made. Rationale distillation (Gupta et al., 2023) incorporates explanations, yet typically lacks mechanisms to ensure the correctness or consistency of those explanations. Concurrent document VQA distillation efforts have explored compiling agentic reasoning pipelines into smaller models, but DocVAL differs by making validation central to the supervision loop and by explicitly correcting spatial grounding errors through deterministic pixel-level feedback. As a result, both reasoning-based supervision & distillation pipelines remain vulnerable to low-quality or misleading training signals.

Validation-driven learning offers a promising direction for addressing these issues. Prior work has explored validation-based feedback in domains such as planning (Verma et al., 2025), but such approaches operate with discrete action spaces and binary correctness signals, limiting their applicability to continuous spatial reasoning. DocVAL adapts validation-driven learning to document spatial grounding by combining textual accuracy, geometric consistency, and reasoning coherence into a rule-based validator that provides deterministic filtering and fine-grained corrective feedback. This formulation enables iterative refinement in continuous spatial domains while avoiding hallucinated supervision and external grounding dependencies.

## 3. Methodology

We present **DocVAL**, a validation-driven reasoning distillation framework for learning reliable spatial grounding in document understanding. Figure 1 provides an overview of the three-phase training and deployment pipeline. Conceptually, DocVAL addresses the following challenge: **"How can a compact vision-language model learn *verifiable, location-aware reasoning* from large teachers, without**

inheriting their inference cost or relying on auxiliary modules at test time?" DocVAL resolves this by **(i)** distilling *explicit spatial reasoning* from a large teacher, **(ii)** rigorously validating and filtering this supervision using a rule-based validator, and **(iii)** iteratively refining a compact student model through validation-driven feedback. We describe each component below.

## 3.1. Overview of the DocVAL Framework

DocVAL operates in three phases:

**Phase A (Validated Teacher Trace Generation).** A large teacher model generates spatial CoT reasoning traces for document-question pairs. These traces are validated and filtered using a rule-based validator (VAL) to ensure correctness, grounding fidelity, and reasoning consistency.

**Phase B (Validation-Driven Student Learning).** A compact student model is trained in two stages: (B1) supervised learning on validated teacher traces, and (B2) iterative refinement using fine-grained corrective feedback produced by VAL.

**Phase C (Pure VLM Deployment).** After training, only the student model is retained. Inference requires a single forward pass over the document image and question, with no OCR, detection, or external validation modules.

Throughout training, auxiliary tools such as text detection are used *only as scaffolding* for validation and supervision. The final model performs end-to-end spatial reasoning directly from visual inputs.

## 3.2. VAL: Rule-Based Validator for Answer Localization

At the core of DocVAL is **VAL**, a rule-based validator designed to enforce quality control over spatial reasoning supervision and to provide structured, actionable feedback for student refinement. VAL plays a *supervisory* role during training: it assesses the correctness of answers, bounding boxes, and reasoning traces, and identifies specific failure modes that require correction.

**Why rule-based validation?** VAL is intentionally not implemented as a learned critic. Our goal is not to approximate human judgment or to learn an implicit notion of correctness, but to provide *stable, verifiable, and reproducible supervision signals* during training. In this setting, a learned validator introduces several challenges: (i) non-deterministic feedback that can vary across training iterations, (ii) susceptibility to hallucinated or inconsistent critiques, (iii) difficulty in providing precise geometric corrections (e.g., pixel-level bounding box adjustments), and (iv) the risk of compounding errors when a learned validator itself is imperfect. By contrast, rule-based validation enables deterministic checks

grounded in explicit metrics (e.g., ANLS, IoU), ensures consistency across iterations, and allows fine-grained, auditable feedback without introducing an additional learning problem. We do not claim that rule-based validation is universally superior; rather, it is well-suited to VAL's role as a training-time assessor providing reliable supervision for spatial grounding. A sensitivity analysis of VAL's thresholds and component weights is provided in Appendix G. *Details about formulation of validation criteria appear in Appendix B.2.*

**Uses of VAL.** Because VAL functions as a validator and assessor rather than a predictive model, we employ it for two complementary purposes within DocVAL:

1. **Training data curation (Phase A):** VAL filters teacher-generated reasoning traces, retaining only high-quality examples for supervised learning.

2. **Validation-driven refinement (Phase B2):** VAL analyzes student predictions and generates fine-grained diagnostic feedback to guide iterative correction.

Both uses rely on the same underlying validation architecture but differ in output granularity, as illustrated in Figure 2. **Appendix D** illustrates this with the help of an example.

**Asymmetric Validation Design.** VAL is permitted to access detected text regions during training-time validation, while the student model *never* receives region-level inputs. This intentional asymmetry allows VAL to perform precise error diagnostics—such as identifying semantic region mismatches (e.g., *Subtotal* vs. *Total*)—while forcing the student to learn spatial grounding purely from visual representations. As a result, region information serves only as temporary supervision scaffolding and is fully removed at inference time.

### 3.2.1. VALIDATION MODULES

VAL consists of five specialized modules:

**Module 1: OCR Grounding Engine.** Matches predicted bounding boxes to detected text regions using IoU, enabling semantic verification (e.g., detecting "Subtotal" vs. "Total" confusions) independent of geometric accuracy.

**Module 2: Answer Validator.** Computes textual correctness using a weighted combination of ANLS and OCR presence:

$$Q_{\text{ans}} = 0.7 \cdot \text{ANLS}(a_p, a_{gt}) + 0.3 \cdot \mathbb{1}[a_p \in \text{OCR}(R)].$$

**Module 3: Bounding Box Validator.** Evaluates spatial accuracy using IoU and region agreement, and computes a

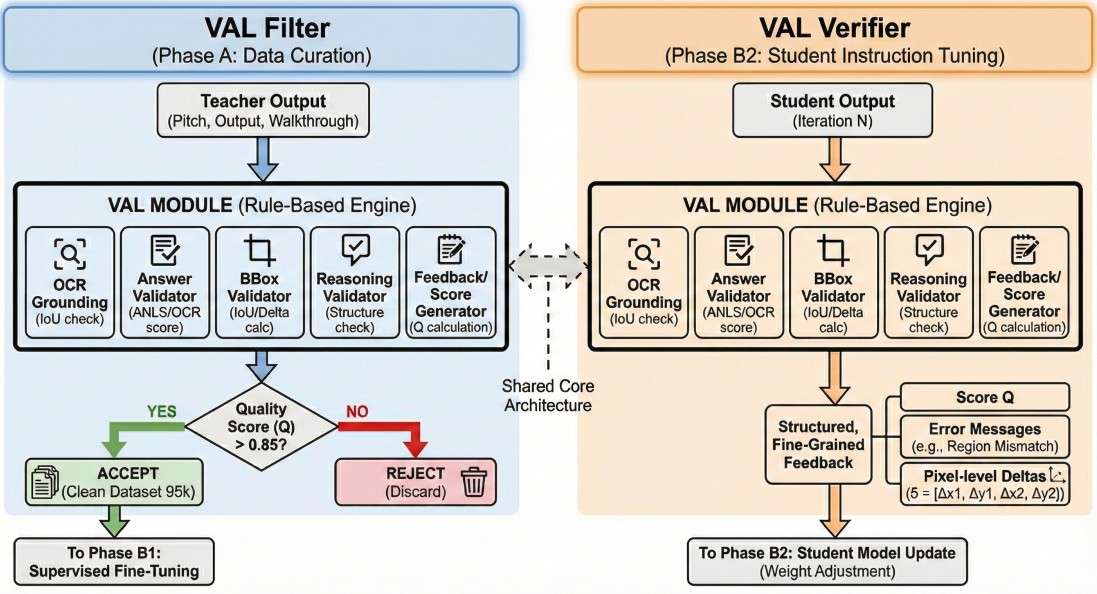

*Figure 2.* **VAL Dual-Mode Architecture Comparison.** VAL operates in two distinct modes sharing the same 5-module architecture but differing in output granularity. **Left (Filter):** Phase A processes teacher outputs ($D_2$) at 50 examples/sec, producing binary Accept/Reject decisions to curate 95K high-quality training examples from 102K raw outputs. **Right (Verifier):** Phase B2 processes student outputs ($D_4$) at 12 examples/sec during iterative refinement, generating detailed rule-based feedback including specific error messages, pixel-level corrections ($\delta$), priority-ordered fixes, and corrected outputs. Both modes compute identical quality scores (Q) but Filter uses Q only for thresholding while Verifier generates comprehensive diagnostic feedback to guide student learning. Refer **Appendix D** for examples.

pixel-level correction vector

$$\delta = [\Delta x_1, \Delta y_1, \Delta x_2, \Delta y_2]^T.$$

**Module 4: Reasoning Validator.** Assesses reasoning quality via three checks: structural completeness, coordinate consistency, and spatial-language alignment.

**Module 5: Feedback Generator.** In filtering mode, outputs binary accept/reject decisions. In verification mode, generates structured natural-language feedback with prioritized corrections and suggested fixes.

### 3.3. Phase A: Validated Teacher Trace Generation

We begin with a dataset $D_1$ of 102,447 document images paired with questions and ground-truth answer bounding boxes. A large teacher VLM (Gemini 2.5 Pro by default) generates spatial CoT traces conditioned on the document image, question, and ground truth annotations.

**Text Detection for Validation.** Text detection (DB-ResNet (Liao et al., 2020)) is applied solely to enable validation. Detected regions allow VAL to verify that predicted boxes correspond to meaningful text and to generate region-level semantic feedback. These regions are *never* used as student inputs.

**Quality-Based Curation.** Each teacher-generated example is scored by VAL. Examples with quality score $Q >$ 0.85 are retained, yielding 95K validated traces (92.7% retention). This curated dataset is split into training, refinement, and test sets (80/10/10). Complete data flow details appear in Appendix D.

### 3.4. Phase B: Validation-Driven Student Learning

#### 3.4.1. STAGE B1: SUPERVISED SPATIAL REASONING LEARNING

The student model (Gemma3-12B) is trained via supervised fine-tuning on the validated dataset. Given only $(I, q)$ as input, the student learns to generate the full output tuple $(CoT, a, b)$, where $b = [x_1, y_1, x_2, y_2]$.

The training objective is standard autoregressive likelihood:

$$\mathcal{L}_{\text{SFT}} = -\sum_{t=1}^{T} \log P_\theta(c_t \mid I, q, c_{<t}),$$

—where $c$ denotes the serialized reasoning, answer, and bounding box tokens.

This stage teaches student to internalize spatial structure & reasoning patterns w/o relying on explicit region anchors.

#### 3.4.2. STAGE B2: ITERATIVE REFINEMENT WITH VAL FEEDBACK

To further improve grounding fidelity, we perform validation-driven instruction tuning. At each iteration, the

student generates predictions, which are evaluated by VAL in verification mode. VAL produces structured natural-language feedback describing errors and pixel-level corrections.

These feedback-augmented examples are added to the training set, and the student is fine-tuned to incorporate the corrections. This process repeats until convergence or a maximum of $\eta_{\max} = 20$ iterations. Unlike prompt-based refinement, this procedure updates model weights, enabling durable improvements in spatial reasoning. A detailed wall-clock and example-pass breakdown comparing Stage B1, Stage B2, and standard distillation is provided in Appendix H.

### 3.5. Phase C: Deployment

At inference time, the trained student model processes a document image and question in a single forward pass, producing reasoning, answer text, and bounding box predictions. No OCR, detection, teacher, or validator is required. This yields a compact, efficient, and verifiable document understanding system suitable for real-world deployment.

## 4. Experiments

Our experimental evaluation is designed to answer three questions:

1. Does DocVAL improve both answer accuracy and spatial grounding compared to existing VLMs?

2. Which components of DocVAL are responsible for these gains?

3. How do data quality, data scale, and model capacity affect spatial reasoning performance?

### 4.1. Datasets and Evaluation Metrics

**Datasets.** We evaluate on five widely used document understanding benchmarks: DocVQA (Mathew et al., 2021), VisualMRC (Tanaka et al., 2021), FUNSD (Jaume et al., 2019), CORD (Park et al., 2019), and SROIE (Huang et al., 2019). Together, these datasets cover diverse document types including scanned forms, receipts, invoices, and web-based documents. Across all datasets, we construct an initial pool of 102,447 examples ($\mathcal{D}_1$). After teacher generation and validation filtering (Section 3.2), 95K examples remain and are split into training ($\mathcal{D}_3$, 76K), refinement ($\mathcal{D}_4$, 9.5K), and held-out test sets ($\mathcal{D}_{\text{test}}$, 9.5K).

**Evaluation Metrics.** We report answer accuracy using ANLS (Li & Liu, 2007). To evaluate spatial grounding, we introduce mean Average Precision (mAP) over bounding boxes, computed across IoU thresholds from 0.5 to 0.95 with

a step size of 0.05. We additionally report IoU@0.5 and IoU@0.75 for interpretability. Since several prior models do not expose localization outputs, mAP is reported only when available. Metric definitions appear in Appendix B.2.

### 4.2. Implementation Details

**Teacher Models.** We evaluate both closed-source and open-source teachers. Closed-source models include Gemini 2.5 Pro (default), GPT-5, Claude 4.5 Sonnet, Gemini 2.5 Flash, and GPT-4o. Open-source teachers include Qwen3-VL-235B-Thinking and Llama4-400B.

**Student Model.** Our primary student is Gemma3-12B, with additional experiments on Gemma3-4B. All students are fully fine-tuned using AdamW. Training is performed on 2×H100 80GB GPUs. Full hyperparameters are provided in Appendix B and Table 5.

**Text Detection.** Text detection is performed using DB-ResNet (Liao et al., 2020) and is used *only* during teacher generation and validation. Alternative detectors (CRAFT (Baek et al., 2019), PSENet (Wang et al., 2019), EasyOCR) are evaluated in ablations. No detection is used during student inference.

**Baselines.** We compare against recent OCR-based and OCR-free document understanding models, including DocLayLLM (Liao et al., 2024), LayoutLLM (Luo et al., 2024), LayTextLLM (Lu et al., 2024), DLaVA (Mohammadshirazi et al., 2024), and ARIAL (Mohammadshirazi et al., 2025a), together with strong general-purpose VLMs such as InternVL3.5, Qwen3-VL, and Llama-Vision variants. These baselines were selected to cover the main competing paradigms for document VQA: OCR-based layout-aware models, OCR-free end-to-end VLMs, document-specific localization systems, and recent general VLMs with strong multimodal reasoning capability. This provides a balanced comparison across models that differ in their use of OCR, layout tokens, localization modules, and model scale. We also include controlled ablations that remove key DocVAL components to isolate the effect of validated reasoning supervision, VAL filtering, and iterative feedback-based refinement.

## 5. Results

### 5.1. Main Results

Table 1 summarizes performance across all benchmarks. DocVAL consistently improves both answer accuracy and spatial grounding, establishing state-of-the-art results among compact, end-to-end, OCR-free-at-inference VLMs, while also outperforming strong modular grounding baselines on overlapping benchmarks (see Appendix F for a detailed breakdown).

*Table 1.* Performance comparison on Document VQA benchmarks. DocVAL operates as pure VLM without text detection at inference. [†]Uses OCR at inference; not directly comparable to compact OCR-free VLMs. Best results per size category in **bold**.

| Method | Model Size | OCR@Inf | DocVQA | | VisualMRC | | FUNSD | | CORD | | SROIE | |
|---|---|---|---|---|---|---|---|---|---|---|---|---|
| | | | ANLS | mAP | ANLS | mAP | ANLS | mAP | ANLS | mAP | ANLS | mAP |
| *Prior Work* | | | | | | | | | | | | |
| DocLayLLM | Llama3-7B | ✗ | 78.4 | - | 55.0 | - | 84.1 | - | 71.3 | - | 84.3 | - |
| LayoutLLM | Vicuna1.5-7B | ✗ | 74.3 | - | 55.8 | - | 80.0 | - | 63.1 | - | 72.1 | - |
| LayTextLLM | Llama2-7B | ✗ | 77.2 | - | 41.7 | - | 81.0 | - | 82.5 | - | 96.1 | - |
| DLaVA | Pixtral-12B | ✗ | 85.9 | 46.2 | 52.1 | - | 87.6 | 45.5 | 84.4 | 57.9 | 91.4 | - |
| ARIAL | Gemma3-27B | ✓ | 88.7 | 50.1 | - | - | 90.0 | 50.3 | 85.5 | 60.2 | 93.1 | - |
| *VLMs (4B-8B)* | | | | | | | | | | | | |
| Qwen3-VL-8B-Thinking | 8B | ✗ | 80.4 | - | 53.8 | - | 84.1 | - | 80.7 | - | 86.3 | - |
| InternVL3.5-8B | 8B | ✗ | 79.9 | - | 52.6 | - | 85.3 | - | 82.9 | - | 90.8 | - |
| Gemma3-4B | 4B | ✗ | 78.6 | - | 51.1 | - | 81.9 | - | 78.3 | - | 87.4 | - |
| **DocVAL (Gemma3-4B)** | **4B** | ✗ | **88.7** | **69.1** | **58.4** | **50.1** | **86.0** | **72.3** | **85.6** | **70.2** | **93.1** | **71.8** |
| *VLMs (11B-14B)* | | | | | | | | | | | | |
| Llama-3.2-11B-Vision | 11B | ✗ | 83.2 | - | 54.4 | - | 86.4 | - | 81.6 | - | 90.7 | - |
| InternVL3.5-14B | 14B | ✗ | 84.7 | - | 57.1 | - | 85.9 | - | 84.1 | - | 91.6 | - |
| Gemma3-12B | 12B | ✗ | 84.6 | - | 56.1 | - | 86.9 | - | 85.3 | - | 92.4 | - |
| **DocVAL (Gemma3-12B)** | **12B** | ✗ | **91.4** | **82.4** | **73.7** | **68.8** | **92.2** | **81.8** | **88.8** | **78.1** | **95.2** | **79.6** |

*Table 2.* Validation strategy ablation across datasets.

| Validation Mode | DocVQA | | VisualMRC | | FUNSD | | CORD | | SROIE | |
|---|---|---|---|---|---|---|---|---|---|---|
| | ANLS | mAP | ANLS | mAP | ANLS | mAP | ANLS | mAP | ANLS | mAP |
| No validation (102K data) | 88.1 | 63.7 | 70.4 | 53.3 | 89.1 | 64.2 | 85.6 | 66.4 | 92.2 | 65.1 |
| VAL Filter only (binary) | 89.5 | 76.1 | 71.6 | 64.7 | 90.3 | 69.9 | 86.9 | 71.3 | 93.4 | 70.9 |
| **VAL Filter & Verifier (detailed)** | **91.4** | **82.4** | **73.7** | **68.8** | **92.2** | **81.8** | **88.8** | **78.1** | **95.2** | **79.6** |

On DocVQA, DocVAL with Gemma3-12B achieves 91.4% ANLS and 82.4% mAP, improving over the base Gemma3-12B by +6.8 ANLS points and introducing strong spatial grounding where prior models do not report mAP. Compared to InternVL3.5-14B (84.7% ANLS), DocVAL improves accuracy by +6.7 points while additionally providing precise localization.

These gains generalize across datasets, including +17.6 mAP on FUNSD and +20.2 mAP on VisualMRC relative to the strongest available baselines. Notably, DocVAL achieves these improvements without relying on OCR or detection at inference time.

ARIAL (Mohammadshirazi et al., 2025a) is the strongest directly comparable grounding-focused baseline, explicitly optimizing answer localization via an agentic modular pipeline with OCR at inference. Despite using a 2.3× larger backbone (Gemma3-27B + TrOCR) and a hard OCR dependency at inference, ARIAL achieves 88.7 ANLS and 50.1 mAP on DocVQA. DocVAL (Gemma3-12B) surpasses this with 91.4 ANLS and 82.4 mAP—a **+32.3 mAP gain**—while operating as a pure VLM with no OCR at inference. This confirms that validated spatial CoT distillation is a more effective path to precise answer localization than OCR-based region matching, even at a fraction of the model size (see Appendix F for the per-benchmark breakdown).

**Efficiency at Smaller Scales.** The 4B DocVAL variant achieves 88.7% ANLS and 69.1% mAP on DocVQA, outperforming larger 8B models by a wide margin. Relative to the base Gemma3-4B, DocVAL improves accuracy by +10.1 ANLS points, demonstrating that validated reasoning transfer can compensate for reduced model capacity.

## 5.2. Ablation Studies

The ablations in Tables 2, 3, and 7 together cover the key standard distillation baselines. Specifically, "w/o teacher CoT" corresponds to direct answer/box supervision; "No validation" to unfiltered teacher-output distillation; and "Phase B1 only" to one-shot rationale distillation without iterative refinement. This allows us to isolate the gains from validated, iterative reasoning transfer rather than from additional supervision alone.

### 5.2.1. TEXT DETECTION ANALYSIS

Text detection is used only during training for teacher generation & validation, and is never provided to the student at inference. As shown in Table 6 & Appendix E.1, removing detection from Phase A causes a substantial drop in localization performance (DocVQA mAP: 82.4 → 74.1, –8.3), confirming that detection-guided teacher reasoning & region-based VAL feedback are critical for curating high-

*Table 3.* Training strategy ablation across datasets.

| Training Config | DocVQA | | VisualMRC | | FUNSD | | CORD | | SROIE | |
|---|---|---|---|---|---|---|---|---|---|---|
| | ANLS | mAP | ANLS | mAP | ANLS | mAP | ANLS | mAP | ANLS | mAP |
| Phase B1 only (no iteration) | 88.3 | 72.7 | 72.2 | 60.4 | 90.1 | 70.8 | 86.1 | 69.2 | 92.3 | 70.4 |
| B1 + B2 (5 iterations) | 88.8 | 74.3 | 72.2 | 62.6 | 89.9 | 74.7 | 86.5 | 72.3 | 93.1 | 73.8 |
| B1 + B2 (10 iterations) | 90.3 | 78.9 | 72.6 | 66.4 | 91.4 | 79.2 | 88.0 | 76.2 | 94.5 | 78.1 |
| **B1 + B2 (converged)** | **91.4** | **82.4** | **73.7** | **68.8** | **92.2** | **81.8** | **88.8** | **78.1** | **95.2** | **79.6** |
| *(iterations)* | *(14)* | *(14)* | *(16)* | *(16)* | *(13)* | *(13)* | *(12)* | *(12)* | *(11)* | *(11)* |

*Table 4.* Data scale ablation across datasets. Performance of DocVAL (Gemma3-12B) trained on varying proportions of the validated CoT dataset $\mathcal{D}_3$. All configurations use full iterative refinement on $\mathcal{D}_4$ (9.5K). Best results in **bold**.

| Data Scale | Size | DocVQA | | VisualMRC | | FUNSD | | CORD | |
|---|---|---|---|---|---|---|---|---|
| | | ANLS | mAP | ANLS | mAP | ANLS | mAP | ANLS | mAP |
| 25% | 19K | 86.2 | 64.8 | 68.4 | 54.2 | 87.8 | 66.3 | 83.7 | 62.1 |
| 50% | 38K | 88.9 | 73.6 | 71.1 | 61.7 | 89.9 | 74.1 | 86.3 | 70.8 |
| 75% | 57K | 90.4 | 78.9 | 72.8 | 65.9 | 91.4 | 78.6 | 87.9 | 75.3 |
| 100% | 76K | **91.4** | **82.4** | **73.7** | **68.8** | **92.2** | **81.8** | **88.8** | **78.1** |

quality supervision. Based on the analysis of the results in Table 6, we use DB-ResNet for all reported experiments, while the final student remains a pure VLM at inference.

### 5.2.2. TEACHER MODEL ANALYSIS

Teacher model choice strongly impacts student grounding quality. As summarized in Table 7 (Appendix E.2), reasoning-capable teachers consistently outperform non-reasoning variants. Gemini 2.5 Pro yields the strongest results (91.4 ANLS, 82.4 mAP on DocVQA), while removing explicit teacher reasoning traces during distillation results in a large localization drop (–17.0 mAP). These results indicate that explicit reasoning traces are essential for transferring spatial grounding, beyond what answer or coordinate supervision alone can provide.

### 5.2.3. VALIDATION STRATEGY ANALYSIS

Table 2 evaluates the impact of validation strategy. Training on unfiltered teacher outputs (102K examples without VAL Filter) yields 88.1 ANLS and 63.7 mAP on DocVQA—18.7 mAP below full DocVAL. This substantial gap demonstrates that low-quality traces containing coordinate hallucinations or reasoning inconsistencies significantly impair learning.

Applying VAL Filter in binary mode improves performance to 89.5 ANLS and 76.1 mAP by removing problematic examples. Further incorporating the VAL Verifier with detailed feedback achieves 91.4 ANLS and 82.4 mAP, adding +6.3 mAP over binary filtering alone. These gains confirm that actionable, pixel-level corrections and priority-ordered fixes enable substantially more effective learning than coarse quality filtering. The consistent improvements across all datasets (4.1–11.9 mAP gains from detailed feedback) vali-

date the effectiveness of the dual-mode VAL architecture.

### 5.2.4. TRAINING STRATEGY

Table 3 analyzes training configurations. Stage B1 only (supervised learning without iteration) achieves 88.3 ANLS and 72.7 mAP on DocVQA—establishing baseline spatial reasoning capabilities but leaving substantial room for improvement. Adding iterative refinement yields progressive gains: 5 iterations reach 74.3 mAP (+1.6), 10 iterations 78.9 mAP (+6.2), and convergence at 14 iterations reaches 82.4 mAP (+9.7).

### 5.2.5. DATA SCALE ANALYSIS

Table 4 examines how training data volume affects DocVAL performance. We train the student model on 25%, 50%, 75%, and 100% of $\mathcal{D}_3$ (corresponding to 19K, 38K, 57K, and 76K validated CoT traces respectively), while maintaining the full iterative refinement procedure on $\mathcal{D}_4$. This setup isolates the impact of supervised learning data quantity from the effects of validation-driven refinement.

The results reveal a consistent positive correlation between data scale and performance across all benchmarks. On DocVQA, scaling from 25% to 100% yields substantial gains: +5.2 ANLS points (86.2→91.4) and +17.6 mAP (64.8→82.4). Notably, spatial grounding (mAP) benefits more dramatically from increased data than textual accuracy (ANLS), suggesting that learning precise coordinate regression requires more diverse training examples than semantic understanding. While the 25% configuration (19K examples) achieves reasonable ANLS (86.2%), it struggles significantly with localization (64.8 mAP), indicating that spatial reasoning requires sufficient training diversity to

generalize effectively.

Cross-dataset analysis reveals distinct scaling dynamics. VisualMRC shows the steepest improvement curve (+14.6 mAP from 25% to 100%), likely because its web-based documents exhibit greater layout diversity that benefits from larger training sets. FUNSD and CORD demonstrate more gradual improvements (+15.5 and +16.0 mAP respectively), suggesting that form-based documents contain more regular spatial patterns that can be learned from fewer examples. Importantly, even at 50% data (38K examples), DocVAL achieves 73.6 mAP on DocVQA—already surpassing the unvalidated baseline's 63.7 mAP (Table 2) trained on the full 102K examples.

Finally, a marginal returns analysis shows diminishing but consistent gains as data increases. The 25%→50% jump provides +8.8 mAP on DocVQA, 50%→75% adds +5.3 mAP, and 75%→100% contributes +3.5 mAP. This pattern indicates that while performance continues to improve with scale, the validated CoT approach extracts substantial value from limited data—a critical property for domains where annotation is expensive. The interaction between data scale and iterative refinement merits further investigation: preliminary experiments suggest that refinement partially compensates for limited initial data, though this effect saturates at very low data scales (<25%).

### 5.3. Discussion

**Validated Reasoning as the Core Driver.** Across all ablations, the largest gains arise from explicit, validated reasoning transfer. The combination of CoT supervision and multi-aspect validation prevents error propagation and enables precise spatial learning.

**Why Pure VLMs Can Work.** Our results challenge the assumption that document understanding requires permanent OCR or layout modules. With appropriate supervision, VLMs can internalize spatial representations sufficient for pixel-level localization.

**Implications Beyond Documents.** The DocVAL paradigm generalizes to tasks requiring grounded and decision-centric reasoning, including referring expressions, embodied perception, robotic manipulation, and high-stakes multimodal forecasting where calibrated evidence and confidence is important (Neogi et al., 2025b). The key insight—validated reasoning transfer—extends beyond document understanding.

**Scope and Limitations.** DocVAL's benchmarks span scanned forms, receipts, invoices, and web documents, but do not fully cover handwriting-heavy pages, severely degraded scans, or chart-dominant layouts where both detector quality and teacher reasoning degrade. Because text detection serves only as training-time scaffolding (the student

is OCR-free at inference), detector failures affect supervision quality rather than the deployed inference pipeline. We expand on this scope and on VAL's robustness to its own design choices in Appendix G and Appendix H.

## 6. Conclusion

We introduced DocVAL, a validated chain-of-thought distillation framework that enables compact vision-language models to perform precise spatial grounding in document understanding tasks. By combining reasoning-focused teacher supervision, rule-based multi-aspect validation, and iterative refinement, DocVAL transfers spatial reasoning capabilities without introducing inference-time dependencies. Our results demonstrate that explicit, quality-controlled reasoning transfer outperforms implicit feature distillation and unfiltered scaling. DocVAL establishes a general template for distilling structured reasoning from large models into efficient, deployable systems, with implications extending beyond documents to grounded vision-language reasoning more broadly. As future work, we plan to extend validation-driven distillation to more complex document reasoning settings, such as multi-hop questions and long-form documents, while exploring hybrid validators that retain the reliability of rule-based supervision with greater flexibility. An orthogonal direction is to combine validated reasoning transfer with model-compression and routing-aware specialization techniques, including expert pruning in mixture-of-experts models, to further reduce inference cost while preserving grounded reasoning quality (Neogi et al., 2025a).

## Impact Statement

This paper presents DocVAL, a validated chain-of-thought distillation framework for grounded Document VQA. By encouraging both answer correctness and spatial consistency during training, DocVAL aims to reduce ungrounded outputs and make document understanding models more verifiable and practical in deployment, including settings where OCR-based pipelines may be brittle or unavailable.

DocVAL may have positive societal impact by supporting human verification and auditability in high-stakes workflows (Neogi et al., 2025c) such as finance, healthcare, and legal document processing, where errors can be costly. At the same time, document understanding models may be misused for privacy-invasive information extraction or relied upon for automated decision-making without appropriate oversight. We encourage responsible deployment with human-in-the-loop review, access controls, and privacy-aware data handling. Overall, we believe the primary impact of this work is to advance reliable and interpretable machine learning for grounded document understanding.

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

# A. Terminology and Design Decisions

**CoT Output.** Throughout the paper, "CoT Output" refers to the complete tuple $(CoT, a, b)$ where $CoT$ is step-by-step spatial reasoning, $a$ is the extracted answer, and $b$ is the predicted bounding box. VAL validates all three components.

**Instruction Tuning.** Stage B2 employs instruction tuning which updates model weights through gradient descent on feedback-augmented examples (student mistakes + VAL feedback $\rightarrow$ corrections). This differs from prompt tuning which freezes weights and only trains soft embeddings. Spatial reasoning requires modifying visual encoders and coordinate regression heads, necessitating full weight updates.

**Validation-Driven Learning.** Stage B2 implements an iterative loop: predict $\rightarrow$ validate $\rightarrow$ feedback $\rightarrow$ retrain. VAL actively guides improvement through actionable corrections rather than passive scoring, creating dynamic training datasets that evolve with model capabilities.

**Text Region.** A detected text instance $r_i = (b_i, t_i, i)$ where $b_i \in \mathbb{R}^4$ are bounding box coordinates, $t_i$ is OCR-extracted text, and $i$ is the region index. Text detection (DB-ResNet) extracts region set $R = \{r_1, ..., r_n\}$ with $n \approx$ 15-20 per document. Regions enable VAL's semantic validation ("Region #7 Subtotal vs Region #2 Total") but are never given to students, who must learn text localization through pure visual reasoning.

**Detection as Scaffolding.** Text detection serves training-time only purposes: (1) guiding teacher CoT generation, (2) enabling VAL's semantic validation. Students receive only $(I, q)$ and must develop internal spatial representations. This asymmetry enables quality supervision while maintaining deployment simplicity.

**Design Rationale.** We use rule-based VAL for: (1) deterministic consistency across iterations, (2) precise pixel-level feedback, (3) zero computational cost, (4) objective metrics that cannot hallucinate, and (5) avoiding validation-of-validator recursion. Iterative refinement enables progressive error correction through feedback targeting actual mistakes. Full fine-tuning rather than LoRA is necessary as spatial reasoning requires modifying core visual encoders (preliminary experiments showed 8-12 mAP degradation with parameter-efficient methods).

**Broader Applicability Across Constrained Domains.** The same principle may be useful in other constrained prediction settings, including biometric recognition, where geometric and texture cues support verification (Guha Neogi, 2020; Kar & Neogi, 2020); secure visual communication, where image transformations must preserve hidden-message recoverability (Neogi et al., 2020); and optimization problems such as VLSI routing, workflow scheduling, and wireless sensor-network routing, where predictions must satisfy resource, latency, or connectivity constraints (Neogi, 2019a;b;c; Nath et al., 2018). It may also benefit time-series prediction systems, where reliable forecasts often require models to account for temporal structure, domain constraints, and interpretable evidence rather than producing black-box predictions (Neogi, 2025; Mohammadshirazi et al., 2025c; Neogi, 2023). These domains differ from document VQA, but they share a common need for models whose outputs can be checked against explicit structural, temporal, or domain-specific constraints.

# B. Experimental Configuration

## B.1. Complete Hyperparameters

**Key Configurations.** Phase A: Teacher (Gemini 2.5 Pro, temp 0.7) generates 102K CoT traces; DB-ResNet detection provides 15-20 regions/image for validation only (never given to student). Stage B1: Student (Gemma3-12B) receives only $(I, q)$ and produces $(CoT, a, b)$; learning rate $2e - 4$, effective batch 128, 3 epochs over $D_3$ (76K). Stage B2; 2 epochs/iteration over $D_4$ (9.5K); convergence at 12-16 iterations (avg 13.2) when $\bar{\Delta} < 0.2$ mAP over 3 iterations. Phase C: Pure VLM inference on consumer GPUs.

Table 5 provides complete configuration across all phases.

## B.2. VAL Scoring Formulas

Complete mathematical formulations for VAL. All formulas use detected regions $R$ for validation but regions are never provided to students.

*Table 5.* Complete hyperparameter configuration for DocVAL

| Parameter | Phase A | Stage B1 | Stage B2 | Phase C |
|---|---|---|---|---|
| *Student Training* | | | | |
| Model | N/A | Gemma3-12B | Gemma3-12B | Gemma3-12B |
| Learning Rate | N/A | $2e-4$ | $2e-4$ | N/A |
| Batch Size (effective) | N/A | 128 | 128 | 1 |
| Sequence Length | N/A | 2048 | 2048 | 2048 |
| Epochs | N/A | 3 | 2/iter | N/A |
| Optimizer | N/A | AdamW | AdamW | N/A |
| Weight Decay | N/A | 0.01 | 0.01 | N/A |
| Warmup Steps | N/A | 500 | 200 | N/A |
| *VAL Configuration* | | | | |
| Mode | Filter (binary) | N/A | Verifier (detailed) | N/A |
| $Q_{\min}$ | 0.85 | 0.85 | 0.85 | N/A |
| Weights ($\alpha_{\text{ans}}, \alpha_{\text{bbox}}, \alpha_{\text{reason}}$) | 0.4, 0.4, 0.2 | - | 0.4, 0.4, 0.2 | N/A |
| Throughput | 50 ex/sec | N/A | 12 ex/sec | N/A |
| Retention | 92.7% (95K) | N/A | N/A | N/A |
| *Data Splits* | | | | |
| $D_3$ (Training) | 76,000 | 76,000 | - | - |
| $D_4$ (Validation) | 9,500 | - | 9,500 | - |
| $D_{\text{test}}$ (Test) | 9,500 | - | - | 9,500 |
| *Computational Resources* | | | | |
| GPU | CPU only (VAL) | H100 80GB | H100 80GB | A100 40GB |
| Memory | 64GB RAM | 78GB | 72GB | 28GB |

**Quality Scores:**

$$Q_{\text{ans}} = 0.7 \cdot \text{ANLS}(a_p, a_{gt}) + 0.3 \cdot \mathbb{1}[a_p \in \text{OCR}(R)] \tag{1}$$

$$Q_{\text{bbox}} = 0.8 \cdot \text{IoU}(b_p, b_{gt}) + 0.2 \cdot \mathbb{1}[r_p = r_{gt}] \tag{2}$$

$$Q_{\text{reason}} = \frac{S_{\text{struct}} + S_{\text{coord}} + S_{\text{spatial}}}{3} \tag{3}$$

$$Q = 0.4 Q_{\text{ans}} + 0.4 Q_{\text{bbox}} + 0.2 Q_{\text{reason}} \tag{4}$$

**Convergence Detection:**

$$\bar{\Delta}^{(k)} = \frac{1}{w} \sum_{j=k-w+1}^{k} (\text{mAP}_j - \text{mAP}_{j-1}) \tag{5}$$

$$\text{Converged} \iff \bar{\Delta}^{(k)} < 0.2 \text{ and } \max_j \Delta_j^{(k)} < 0.4 \tag{6}$$

where $w = 3$ iterations.

### B.3. Complete Algorithm

Algorithm 1 provides complete pseudocode with three key design principles: (1) Detection for validation only (line 9), (2) Student receives only $(I, q)$ without regions (line 35), (3) VAL uses regions for feedback but student learns visually (lines 46-49).

---

**Algorithm 1** DocVAL: Validated CoT Distillation

---

1: **Input:** $D_1 = \{(I_i, q_i, a_i^{\text{gt}}, b_i^{\text{gt}})\}_{i=1}^{102447}$, Teacher $T$, Detector $D$, Student $M_{\theta_0}$
2: **Output:** Fine-tuned student $M_{\theta*}$
3: **// Phase A: Teacher Generation + VAL Filter**
4: **for** each $(I_i, q_i, a_i^{\text{gt}}, b_i^{\text{gt}}) \in D_1$ **do**
5:      $R_i \leftarrow D(I_i)$                                                      ▷ Detection for validation only
6:      $(\text{CoT}_i^T, a_i^T, b_i^T) \leftarrow T(I_i, R_i, q_i, a_i^{\text{gt}}, b_i^{\text{gt}})$
7:      **if** VAL-Filter$(I_i, R_i, \text{CoT}_i^T, a_i^T, b_i^T, a_i^{\text{gt}}, b_i^{\text{gt}}) > 0.85$ **then**
8:          $D_{\text{filtered}} \leftarrow D_{\text{filtered}} \cup \{(I_i, R_i, q_i, \text{CoT}_i^T, a_i^T, b_i^T)\}$
9:      **end if**
10: **end for**
11: Split $D_{\text{filtered}} \rightarrow D_3$ (76K), $D_4$ (9.5K), $D_{\text{test}}$ (9.5K)
12: **// Stage B1: Supervised Learning**
13: **for** epoch $e = 1$ to 3 **do**
14:      **for** each $(I_i, q_i, \text{CoT}_i^{\text{gt}}, a_i^{\text{gt}}, b_i^{\text{gt}}) \in D_3$ **do**
15:          **// CRITICAL: Student gets only $(I_i, q_i)$, no regions**
16:          $\mathcal{L} \leftarrow -\sum_t \log P_\theta(c_t | I_i, q_i, c_{<t})$ where $c = [\text{CoT}, a, b]$
17:          Update $\theta$ with AdamW
18:      **end for**
19: **end for**
20: **// Stage B2: Iterative Refinement**
21: **for** iteration $k = 1$ to 20 **do**
22:      **for** each $(I_i, R_i, q_i, a_i^{\text{gt}}, b_i^{\text{gt}}) \in D_4$ **do**
23:          $(c_i^{(k)}, a_i^{(k)}, b_i^{(k)}) \leftarrow M_{\theta_{k-1}}(I_i, q_i)$                              ▷ No regions to student
24:          $F_i \leftarrow$ VAL-Verifier$(I_i, R_i, c_i^{(k)}, a_i^{(k)}, b_i^{(k)}, a_i^{\text{gt}}, b_i^{\text{gt}})$
25:          $D_{\text{corr}}^{(k)} \leftarrow D_{\text{corr}}^{(k)} \cup \{(I_i, q_i, F_i, c_i^{\text{gt}}, a_i^{\text{gt}}, b_i^{\text{gt}})\}$
26:      **end for**
27:      Fine-tune $\theta$ on $D_{\text{corr}}^{(k)}$ for 2 epochs
28:      Evaluate: $\text{mAP}_k$ on $D_4$
29:      **if** Converged (Eq. 6) **then**
30:          **break**
31:      **end if**
32: **end for**
33: **return** $M_{\theta*}$                                                                              ▷ Pure VLM

---

## C. Qualitative Examples

**Success: Complex Receipt.** Multiple similar amounts (Subtotal: \$87.50, Tax: \$7.88, Total: \$108.51). Student correctly distinguished "TOTAL" field, predicted [542,876,618,904]. Result: ANLS=1.0, IoU=1.0 (perfect). Demonstrates semantic understanding and pixel-accurate localization.

**Failure: Multi-Line Address.** Three-line address spanning disconnected regions. Student correctly extracted full text (ANLS=1.0) but bbox covered only first line (IoU=0.31). Single-bbox architectural limitation, not reasoning failure.

**Failure: Handwritten Text.** Student read handwriting correctly (ANLS=1.0) but bbox imprecise (IoU=0.68 vs. typical $> 0.9$ for print). VAL validation during training missed handwritten regions, affecting feedback quality.

**Near-Perfect: Structured Form.** Tax form with clear grid layout. Student identified "EIN" label and predicted [384,187,496,214] vs. GT [385,188,495,213]. ANLS=1.0, IoU=0.98 (1-2px offset). Represents typical performance on well-structured documents where DocVAL excels.

# D. Sample Outputs

## D.1. Teacher, Student, and VAL Interaction

**Teacher Generation (Phase A):**

*Input:* Receipt image, Q: "What is the total?"

*Regions (for teacher/VAL):* 15 detected, including Region #2 [510,800,570,830] → "$45.99", Region #14 [445,795,505,825] → "TOTAL:"

*Teacher Output:* Step-by-step reasoning referencing regions → Answer: $45.99, BBox: [510,800,570,830]

*VAL Filter:* $Q = 1.0 > 0.85 \rightarrow$ ACCEPT

---

**Student Prediction (Iteration 3, Stage B2):**

*Input (Pure VLM):* Same receipt image, Q: "What is the total?" **(NO regions)**

*Output:* Steps 1-5 (identifies "SUBTOTAL" instead of "TOTAL") → Answer: $42.50, BBox: [760,650,840,680]

---

**VAL Verifier Feedback:**

**Validation:** Answer: ANLS=0.167, BBox: IoU=0.0, Reasoning: 0.73, Overall: Q=0.313 → INVALID

**Detailed Feedback:**

- **Answer Error:** Got "$42.50" (SUBTOTAL), Expected "$45.99" (TOTAL). Wrong semantic field.

- **BBox Error:** IoU=0.0. Move 250px LEFT, 150px DOWN. Targets Subtotal (middle) instead of Total (lower).

- **Reasoning Issue:** Step 3 identified "SUBTOTAL" when question asks for "total". Look for "TOTAL" label specifically.

- **Priority Fixes:** (1) Distinguish Subtotal vs Total fields, (2) Locate "TOTAL" label in lower section, (3) Adjust bbox position

# E. Ablation Studies Experiments

## E.1. Text Detection Analysis

Table 6 analyzes the role of text detection during teacher data generation and VAL validation. Removing detection from Phase A (teacher generation and validation filtering) significantly degrades performance: DocVQA mAP drops from 82.4 to 74.1 (–8.3). This confirms that detection-guided teacher CoT generation and region-based VAL feedback are critical for maintaining high training data quality. Notably, the student model operates without detection at inference, demonstrating that high-quality supervision—rather than inference-time dependencies—enables pure VLM spatial reasoning.

Among detection methods, DB-ResNet achieves the best overall performance (82.4 mAP on DocVQA), likely due to its strong recall for small text regions and efficient processing. CRAFT (80.6 mAP) and PSENet (79.3 mAP) show slightly lower performance, while EasyOCR (78.7 mAP) struggles with dense text layouts. Removing region references from teacher CoT while retaining detection for VAL (76.8 mAP) further shows that region-aware reasoning generation contributes beyond validation alone, highlighting detection's dual role as both supervision scaffolding and reasoning guide.

*Table 6.* Text detection ablation across datasets. Detection used only in Phase A (teacher+VAL), not student inference.

| Configuration | DocVQA | | VisualMRC | | FUNSD | | CORD | | SROIE | |
|---|---|---|---|---|---|---|---|---|---|---|
| | ANLS | mAP | ANLS | mAP | ANLS | mAP | ANLS | mAP | ANLS | mAP |
| **DocVAL (DB-ResNet)** | **91.4** | **82.4** | **73.7** | **68.8** | **92.2** | **81.8** | **88.8** | **78.1** | **95.2** | **79.6** |
| DocVAL (CRAFT) | 90.9 | 80.6 | 73.2 | 67.3 | 91.6 | 79.9 | 88.1 | 76.4 | 94.8 | 78.1 |
| DocVAL (PSENet) | 90.5 | 79.3 | 72.8 | 66.1 | 91.1 | 78.5 | 87.7 | 75.2 | 94.4 | 77.3 |
| DocVAL (EasyOCR) | 90.1 | 78.7 | 72.3 | 65.4 | 90.8 | 77.8 | 87.2 | 74.6 | 94.1 | 76.8 |
| w/o detection (Phase A) | 88.7 | 74.1 | 70.9 | 62.2 | 89.3 | 73.4 | 85.9 | 71.7 | 92.6 | 73.9 |
| w/o region refs in teacher CoT | 89.8 | 76.8 | 71.8 | 64.1 | 90.4 | 76.2 | 87.0 | 73.9 | 93.5 | 75.4 |

### E.2. Teacher Model Analysis

Table 7 examines how teacher model choice influences student performance. Gemini 2.5 Pro achieves the strongest results (91.4 ANLS, 82.4 mAP on DocVQA), producing high-quality reasoning traces that transfer effectively to the student. Reasoning-focused models (Claude 4.5 Sonnet: 90.8/81.2; GPT-5: 90.3/80.5) consistently outperform non-reasoning variants (Gemini 2.5 Flash: 88.4/76.1; GPT-4o: 87.9/74.8), confirming that explicit step-by-step reasoning in teacher outputs is crucial for learning spatial grounding.

Open-source teachers achieve competitive performance as well: Qwen3-VL-235B (90.6/79.8) and Llama4-400B (90.1/78.9) generate sufficiently high-quality CoT traces for effective distillation. In contrast, the *w/o teacher CoT* baseline (87.7 ANLS, 65.4 mAP), which uses direct bounding box supervision without reasoning traces, exhibits a 17.0 mAP degradation. This gap confirms that explicit reasoning transfer—not just answer or coordinate supervision—is essential for spatial grounding.

*Table 7.* Teacher model ablation. Student (Gemma3-12B) performance across datasets with different teacher VLMs.

| Teacher Model | DocVQA | | VisualMRC | | FUNSD | | CORD | | SROIE | |
|---|---|---|---|---|---|---|---|---|---|---|
| | ANLS | mAP | ANLS | mAP | ANLS | mAP | ANLS | mAP | ANLS | mAP |
| *Closed-source (reasoning)* | | | | | | | | | | |
| **Gemini 2.5 Pro** | **91.4** | **82.4** | **73.7** | **68.8** | **92.2** | **81.8** | **88.8** | **78.1** | **95.2** | **79.6** |
| Claude 4.5 Sonnet | 90.8 | 81.2 | 73.1 | 67.9 | 91.7 | 80.6 | 88.2 | 77.3 | 94.8 | 78.9 |
| GPT-5 | 90.3 | 80.5 | 72.6 | 67.2 | 91.3 | 79.8 | 87.7 | 76.7 | 94.4 | 78.2 |
| *Closed-source (non-reasoning)* | | | | | | | | | | |
| Gemini 2.5 Flash | 88.4 | 76.1 | 71.2 | 64.3 | 89.9 | 76.7 | 86.3 | 73.9 | 93.3 | 75.8 |
| GPT-4o | 87.9 | 74.8 | 70.8 | 63.1 | 89.4 | 75.2 | 85.8 | 72.6 | 92.9 | 74.4 |
| *Open-source* | | | | | | | | | | |
| Qwen3-VL-235B-A22B | 90.6 | 79.8 | 72.8 | 66.5 | 91.4 | 79.1 | 88.0 | 76.2 | 94.6 | 77.8 |
| Llama4-400B-A17B | 90.1 | 78.9 | 72.3 | 65.8 | 90.9 | 78.3 | 87.5 | 75.4 | 94.2 | 77.1 |
| w/o teacher CoT (direct FT) | 87.7 | 65.4 | 68.1 | 58.2 | 88.2 | 68.9 | 86.4 | 68.7 | 92.7 | 71.3 |

## F. ARIAL Comparison: Detailed Breakdown

Table 1 includes ARIAL (Mohammadshirazi et al., 2025a) as a grounding-focused baseline. Here we provide the per-benchmark breakdown of the gap between DocVAL and ARIAL. ARIAL is an agentic modular pipeline combining TrOCR for OCR-based text extraction, a retrieval module, a fine-tuned Gemma3-27B QA backbone, and a separate text-to-region alignment module for localization—making OCR a hard inference-time dependency.

The mAP gap is particularly large because ARIAL grounds via OCR text-region matching, which is bounded by detection coverage and string-match precision, whereas DocVAL trains the student to perform spatial reasoning directly from visual inputs via validated CoT distillation. Improvements to ARIAL's components do not directly address this architectural limitation.

*Table 8.* Per-benchmark comparison between DocVAL (Gemma3-12B, OCR-free) and ARIAL (Gemma3-27B + TrOCR). DocVAL outperforms ARIAL on every overlapping benchmark in both ANLS and mAP, using a 2.3× smaller backbone and no OCR at inference.

| Method | DocVQA | | FUNSD | | CORD | | SROIE |
|---|---|---|---|---|---|---|---|
| | ANLS | mAP | ANLS | mAP | ANLS | mAP | ANLS |
| ARIAL (27B+OCR) | 88.7 | 50.1 | 90.0 | 50.3 | 85.5 | 60.2 | 93.1 |
| DocVAL (12B) | **91.4** | **82.4** | **92.2** | **81.8** | **88.8** | **78.1** | **95.2** |
| Δ | +2.7 | **+32.3** | +2.2 | **+31.5** | +3.3 | **+17.9** | +2.1 |

### F.1. Conceptual Positioning Relative to DOGR and TRIG

Direct numerical comparison with DOGR and TRIG is not feasible because they address related but fundamentally distinct problems. DOGR targets document grounding and referring—localizing a referenced region given a referring expression—evaluated on DOGR-Bench (charts, posters, PDFs), not on the DocVQA/FUNSD/CORD/SROIE benchmarks used here. Its input/output specification differs from DocVAL's: DOGR takes a region description and outputs a localized region, whereas DocVAL takes a natural-language question and outputs both an answer and its bounding box. TRIG targets text-rich image grounding across broader document types and evaluates on TRIG-Bench with different annotation protocols, without reporting mAP@IoU[0.5:0.95] on our benchmarks.

*Table 9.* Conceptual comparison: DocVAL vs. DOGR vs. TRIG.

| Dimension | DocVAL | DOGR | TRIG |
|---|---|---|---|
| Primary task | Answer localization in DocVQA | Document referring | Text-rich grounding |
| OCR at inference | ✗ | Not specified | OCR-assisted |
| Pixel-level iterative refinement | ✓ | ✗ | ✗ |
| Inference-time independence | ✓ | ✗ | ✗ |
| Validated CoT distillation | ✓ | ✗ | ✗ |

Neither DOGR nor TRIG targets the specific combination of validated spatial CoT distillation, OCR-free inference, and compact deployable student training that defines DocVAL. The released 95K validator-verified CoT dataset may also benefit future work in referring and grounding settings.

## G. VAL Sensitivity Analysis

A natural concern with a rule-based validator is whether performance depends sensitively on the specific thresholds and weights chosen in Appendix B.2. We perform two targeted sensitivity studies on DocVQA to test this.

### G.1. Threshold Sensitivity

We vary the global VAL quality threshold $Q_{min}$ from 0.75 to 0.95, holding all other settings fixed. Table 10 reports the corresponding retention rate, resulting dataset size, and student performance.

*Table 10.* Sensitivity of DocVAL (Gemma3-12B) to the VAL quality threshold $Q_{min}$ on DocVQA. Performance is stable across a broad range (0.80–0.90); degradation appears only at the extremes.

| $Q_{min}$ | Retention | Dataset Size | ANLS | mAP |
|---|---|---|---|---|
| 0.75 | 98.1% | ∼100K | 90.1 | 79.3 |
| 0.80 | 96.2% | ∼98K | 90.6 | 80.8 |
| **0.85 (ours)** | **92.7%** | **95K** | **91.4** | **82.4** |
| 0.90 | 83.4% | ∼85K | 90.8 | 81.6 |
| 0.95 | 65.1% | ∼67K | 89.2 | 78.9 |

Across the operational range (0.80–0.90), DocVQA mAP varies by only 1.6 points. Performance degrades only at the extremes—either retaining too many noisy traces (0.75) or discarding too much useful data (0.95). This confirms that $Q_{min} = 0.85$ is not a knife-edge optimum.

## G.2. Component Weight Sensitivity

We next vary the component weights $(\alpha_{\text{ans}}, \alpha_{\text{bbox}}, \alpha_{\text{reason}})$ in the overall quality score (Eq. 4) around the default $(0.4, 0.4, 0.2)$ setting.

*Table 11.* Sensitivity of DocVAL (Gemma3-12B) to the VAL component weights on DocVQA. Maximum mAP variation across all tested configurations is 1.2 points.

| $(\alpha_{\text{ans}}, \alpha_{\text{bbox}}, \alpha_{\text{reason}})$ | ANLS | mAP |
|---|---|---|
| $(0.5, 0.3, 0.2)$ | 91.1 | 81.7 |
| $(0.3, 0.5, 0.2)$ | 91.0 | 82.1 |
| **$(0.4, 0.4, 0.2)$ — ours** | **91.4** | **82.4** |
| $(0.4, 0.3, 0.3)$ | 91.2 | 81.9 |
| $(0.3, 0.3, 0.4)$ | 90.8 | 81.2 |
| $(0.33, 0.33, 0.33)$ | 91.0 | 81.5 |

Across six weight configurations, mAP varies by at most 1.2 points. The default $(0.4, 0.4, 0.2)$ encodes a deliberate priority—answer correctness and box quality as primary, reasoning quality as supporting supervision—but performance is not brittle to this choice. We attribute the stability to two factors: (i) low-quality traces typically fail on multiple dimensions simultaneously, so the rejection decision is robust to weight perturbations; and (ii) VAL's geometric correction vector $\delta$ is computed independently of these scalar weights, so verifier-mode pixel-level feedback is unaffected.

# H. Compute Cost and Scope

## H.1. Stage B2 Compute Breakdown

Stage B2 (iterative refinement) is sometimes perceived as adding substantial training overhead beyond Stage B1. We clarify the actual cost here. Stage B1 trains for 3 epochs over $D_3$ (76K examples), giving $\sim$228K example-passes. Stage B2 runs for 2 epochs per iteration over $D_4$ (9.5K examples) and converges in 12–16 iterations on average (13.2), giving $\sim$251K example-passes total. The two stages are therefore of the same order in example-passes; Stage B2 is not an additional full-scale training run.

VAL verification itself is CPU-only and runs at 12 examples/sec in verifier mode, which is negligible relative to GPU student training. The total Stage B2 wall-clock cost on our 2×H100 setup is roughly 1.0–1.2× that of Stage B1. Importantly, this is a one-time training cost: the deployed student is a pure VLM with no OCR, detector, teacher, or validator at inference.

## H.2. Scope of Current Benchmarks

DocVAL is evaluated on DocVQA, VisualMRC, FUNSD, CORD, and SROIE, covering scanned forms, receipts, invoices, and web documents. We do not claim full coverage of all document regimes. In particular, three regimes lie outside the current benchmark distribution and are expected to be harder:

- **Handwriting-heavy pages.** As noted in Appendix C, our qualitative analysis shows the student can read handwritten text correctly (ANLS = 1.0 in one example) but localizes less precisely (IoU = 0.68 vs. >0.9 for print). This degradation traces back to training-time validation: detectors miss handwritten regions, so VAL's region-level feedback is weaker.

- **Severely degraded scans.** Heavy noise, faded ink, or skewed pages reduce detector recall, which in turn weakens VAL's supervision signal.

- **Chart- and figure-dominant pages.** Documents where the answer-bearing region is a non-text visual element (e.g., a chart axis label inside a plot) fall outside the validator's text-detection scaffolding.

In all three cases, the failure mode affects the *quality of training supervision*, not the deployed inference pipeline—the student receives only $(I, q)$ at test time and is not affected by detector failures at inference. Extending DocVAL to these regimes (e.g., via complementary chart/figure detectors during training, or via teacher CoTs explicitly trained on handwritten data) is an interesting direction for future work.

