# OpenReview forum: "DocVAL: Validated Chain-of-Thought Distillation for Grounded Document VQA"
_ICML.cc/2026/Conference — ICML 2026 regular_

### Official Review · Reviewer_B4eX · 2026-03-10

**Soundness:** 3
**Presentation:** 3
**Significance:** 3
**Originality:** 2
**Overall Recommendation:** 4
**Confidence:** 4

**Summary:**

The paper proposes DocVAL, a training framework for document question answering with spatial grounding using vision language models. The approach distills spatial reasoning from large teacher models into smaller student models by generating chain of thought reasoning traces for document questions and filtering them with rule based validation. The validated reasoning data is then used to fine tune a compact vision language model and further refined through validation driven feedback. Experiments on several document understanding benchmarks show improvements in answer accuracy and bounding box localization while the final model performs inference without relying on OCR or external detection modules.

**Compliance With Llm Reviewing Policy:**

Affirmed.

**Final Justification:**

Thank you for the detailed response and for further clarifying the robustness of the validation mechanism. The additional analyses and explanations have addressed my previous concerns regarding the sensitivity of the rule-based design and its applicability across different document settings. The empirical evidence provided in the rebuttal strengthens my confidence in the generality of the proposed framework.

Overall, my main concerns have been satisfactorily resolved. As a result, I am increasing my score from 3 to 4.

**Key Questions For Authors:**

- The motivation connects model scale, spatial grounding ability, and OCR dependence, but the causal relationship between these factors is not clearly established. Could the authors clarify why the performance gap between large and small VLMs should primarily be attributed to the lack of spatial reasoning supervision rather than model capacity or pretraining data?
- The proposed framework is presented as a distillation method. How does DocVAL compare to standard distillation baselines, such as direct teacher output distillation or supervision using answers and bounding boxes without reasoning traces?
- The validation module relies on rule based checks and metrics to filter teacher outputs and provide feedback. How sensitive are the results to the specific validation rules or thresholds used in the VAL module?

**Limitations:**

Yes.

**Strengths And Weaknesses:**

# Strengths:
- The work evaluates the approach on several widely used document benchmarks, including DocVQA, FUNSD, CORD, VisualMRC, and SROIE, providing reasonably broad empirical coverage.
- The method trains a model that performs inference without relying on OCR or external detection modules, which simplifies deployment compared to many existing document pipelines.
- The paper introduces a relatively large dataset of validated reasoning traces aligned with answers and bounding boxes, which could be useful for future work on document reasoning and grounding.

# Weaknesses
- The motivation is not clearly articulated. The introduction mixes several different issues, including model scale, deployment constraints, spatial grounding ability, and OCR dependence, without clearly explaining how these factors are causally related.
- The proposed training framework largely resembles existing chain of thought distillation and validation driven training approaches, and the main contribution appears to be adapting these ideas to document grounding rather than introducing a fundamentally new method.
- The rule based validation module mainly combines standard metrics and heuristic checks to filter teacher outputs and provide feedback. As a result, the validator functions more as an engineering component than a novel learning mechanism.
- The paper presents the approach as a distillation method, but the experiments do not compare against standard distillation baselines or alternative supervision strategies, making it difficult to assess whether the gains come from the proposed framework or from additional supervision and data filtering.
- The strongest contribution of the work appears to be the curated dataset of validated reasoning traces and the associated engineering pipeline, while the methodological novelty of the training framework itself is limited.

---

> ### Author Rebuttal · Authors · 2026-03-29
>
> We thank the reviewer for the careful reading & for recognizing the paper’s broad benchmark coverage, OCR-free inference, & the value of the released 95K validated-CoT dataset. We respectfully disagree, however, that the contribution is mainly an engineering pipeline, & address the main concerns below.
>
> **On the motivation & causal link between model scale, grounding, & OCR dependence:** We agree the introduction should articulate this more clearly. Our claim is not that the large/small gap is caused only by model capacity. Rather, current training paradigms transfer answers or distributions, but do not explicitly supervise how localization decisions are made. Large VLMs can often recover this implicitly from scale & pretraining, while compact VLMs degrade sharply when precise grounding is required. The ablations directly support this: with the same student model capacity, removing teacher CoT (“w/o teacher CoT”) drops DocVQA mAP to 65.4; using raw unvalidated teacher traces gives only 63.7; binary validation raises this to 76.1; & full Filter+Verifier reaches 82.4. These controlled comparisons support the claim that the missing ingredient is not simply more parameters, but high-quality spatial reasoning supervision.
>
> **On whether DocVAL is just standard CoT distillation / validation-driven training adapted to documents:** The key novelty is not generic CoT-to-student transfer. The core contribution is showing that for grounded document QA, naïvely using teacher reasoning is insufficient & can be actively harmful unless the reasoning is validated, filtered, & iteratively corrected. The framework’s main contribution is therefore the combination of: 1) validated spatial CoT distillation, 2) a dual-mode rule-based validator used for both curation & pixel-level corrective feedback, 3) an asymmetric training/inference design where detection is used only as training-time scaffolding but removed entirely at inference.
>
> This is different from one-shot rationale distillation because the contribution is precisely the mechanism that makes reasoning transfer work for continuous spatial grounding.
>
> **On the absence of standard distillation / alternative-supervision baselines:** We agree the paper should frame these baselines more explicitly. The current ablations already cover the key alternatives:
>
> w/o teacher CoT = direct supervision using answer + bounding box without reasoning traces; this is the closest baseline to standard direct supervision / coordinate regression, & it suffers a 17.0 mAP drop.
>  - No validation (102K) = direct teacher-output distillation without validation/filtering, giving only 63.7 mAP.
>  - VAL Filter only = filtered reasoning supervision without verifier-stage refinement, giving 76.1 mAP.
>  - Phase B1 only = supervised learning on validated traces without iterative refinement, giving 72.7 mAP.
>
> So the paper already isolates the reviewer’s question: do the gains come from “more supervision” alone, or from the proposed framework? The answer is that neither raw teacher outputs nor answer/box supervision alone is sufficient; the gains come from validated reasoning transfer plus iterative corrective feedback. We will revise the paper to make this baseline framing clearer.
>
> **On the rule-based validator being “just an engineering component”**
> We agree VAL is deliberately engineered rather than learned. This is intentional. Its role is not to be a new learned optimizer, but to provide deterministic, auditable, pixel-level training-time supervision. In this setting, that is a methodological choice, not a weakness: a learned critic would introduce hallucinated critiques, instability across iterations, & weaker geometric correction ability. VAL’s importance is also empirically demonstrated: moving from no validation to Filter+Verifier improves DocVQA mAP from 63.7 to 82.4. So even if rule-based, it is not a minor implementation detail; it is the core mechanism that enables reliable grounded distillation.
>
> **On sensitivity to validation rules / thresholds:** This is a fair concern. The paper reports the full equations & thresholds in Appendix B.2.
>
> Two points matter here. 1) the same fixed rules & weights are used unchanged across five datasets, multiple detectors, & multiple teacher models, yet the method remains consistently strong. 2) the threshold acts as a high-pass filter rejecting clearly bad examples; in this regime, moderate weight perturbations are less brittle because poor examples typically fail on multiple dimensions simultaneously. We agree that an explicit sensitivity study would strengthen the paper, & we will add this discussion more clearly in the revision.
>
> Overall, we really appreciate the reviewer’s feedback. In the revised version we will make sure to clarify the role of explicit spatial-reasoning supervision, make the existing baseline comparisons more explicit, and better highlight VAL as a deterministic training-time supervision mechanism supported by strong ablations.

---

> > ### Author Rebuttal · Reviewer_B4eX · 2026-04-01
> >
> > Thank you for the detailed response and for clarifying the role of validated spatial reasoning supervision. The ablations demonstrating the impact of validation and iterative correction are helpful and strengthen the empirical support for the proposed framework.
> >
> > However, some concerns remain regarding the generality and robustness of the rule-based validation mechanism. While the same rules are reported to work across datasets, the method still relies on carefully designed geometric constraints and thresholds tailored to document grounding tasks. It remains unclear how sensitive performance is to these design choices and how easily the approach would transfer to settings with different document structures or layouts.

---

> > > ### Author Response · Authors · 2026-04-02
> > >
> > > Thank you for the follow-up. We agree this is an important point, and we examined the remaining concern along the two dimensions you highlight: (1) sensitivity to VAL’s design choices (thresholds and weights) and (2) transferability across document structures/layouts.
> > >
> > > **1. Sensitivity to thresholds and weights:** We ran two targeted sensitivity studies. First, we varied the global VAL quality threshold from 0.75 to 0.95. The main result is that performance is stable across a broad range (0.80–0.90) rather than concentrated at a knife-edge optimum: DocVQA mAP varies by only 1.6 points across this range, with noticeable degradation only at the extremes, where either too many noisy examples are retained (0.75) or too much useful data is discarded (0.95). Second, we varied the component weights around the default setting. Here again, performance remains stable: across all tested weight configurations, the maximum mAP variation is only 1.2 points. These experiments support our original intuition that VAL behaves as a high-pass filter, not a brittle scoring function. Low-quality traces typically fail on multiple dimensions simultaneously, while strong traces remain strong under moderate perturbations.
> > >
> > > **2. Transfer across different document structures and layouts:** We also want to clarify that VAL is built from document-agnostic primitives, not template-specific rules. Its checks are based on answer similarity, geometric overlap, semantic region agreement through detected text, and reasoning consistency / spatial-language alignment. These do not encode assumptions about a specific benchmark layout or field template. Consistent with this, we apply the same rules, weights, and threshold unchanged across all five datasets, multiple teachers, and multiple detectors, and still obtain strong gains throughout. We view this as meaningful empirical evidence that VAL is not narrowly overfit to one document structure.
> > >
> > > At the same time, we agree that broader transfer is not guaranteed automatically. The current benchmarks already span heterogeneous structures—scanned forms, receipts, invoices, and web documents—but there remain document regimes outside this distribution (e.g., handwritten-heavy pages, chart-dominant pages, technical layouts) where both detection quality and teacher reasoning would become harder. We will make this scope clearer in the revision. In that sense, VAL is best understood as a general, transparent, and configurable validation framework for grounded DocVQA — one that already demonstrates strong cross-dataset robustness without per-dataset retuning, while remaining reasonably adaptable to new document domains as needed.
> > >
> > > To make this concrete, we include the following threshold- and weight-sensitivity results/tables in the appendix of the revised version:
> > >
> > > | Q Threshold | Retention Rate | Dataset Size | DocVQA ANLS | DocVQA mAP |
> > > | ----------- | -------------: | -----------: | ----------: | ---------: |
> > > | 0.75        |          98.1% |        ~100K |        90.1 |       79.3 |
> > > | 0.80        |          96.2% |         ~98K |        90.6 |       80.8 |
> > > | 0.85 (ours) |          92.7% |          95K |        91.4 |       82.4 |
> > > | 0.90        |          83.4% |         ~85K |        90.8 |       81.6 |
> > > | 0.95        |          65.1% |         ~67K |        89.2 |       78.9 |
> > >
> > >
> > > | Weight Config `(alpha_ans, alpha_bbox, alpha_reason)` | DocVQA ANLS | DocVQA mAP |
> > > | ----------------------------------------------------- | ----------: | ---------: |
> > > | (0.5, 0.3, 0.2)                                       |        91.1 |       81.7 |
> > > | (0.3, 0.5, 0.2)                                       |        91.0 |       82.1 |
> > > | (0.4, 0.4, 0.2) — ours                                |        91.4 |       82.4 |
> > > | (0.4, 0.3, 0.3)                                       |        91.2 |       81.9 |
> > > | (0.3, 0.3, 0.4)                                       |        90.8 |       81.2 |
> > > | (0.33, 0.33, 0.33)                                    |        91.0 |       81.5 |
> > >
> > >
> > > We hope these results directly address the remaining concern, and we respectfully ask the reviewer to consider raising their score in light of this additional evidence. We remain happy to provide further clarifications during the discussion period.

---

### Official Review · Reviewer_1ytz · 2026-03-11

**Soundness:** 2
**Presentation:** 3
**Significance:** 3
**Originality:** 2
**Overall Recommendation:** 3
**Confidence:** 4

**Summary:**

This paper presents the concept of validated chain-of-thought distillation for grounded Document VQA: a large teacher model produces spatial CoT traces, a rule-based validator (VAL) filters and critiques them, and a compact student is then trained in two stages—supervised fine-tuning on validated traces followed by iterative refinement using structured validator feedback. The final student is a pure VLM at inference time and does not require OCR or detection.

**Compliance With Llm Reviewing Policy:**

Affirmed.

**Key Questions For Authors:**

Your strongest novelty claim is validated CoT distillation for grounded document QA. Why are DOGR and TRIG not discussed more centrally as adjacent grounding-oriented work?

Can you provide a more rigorous justification for mAP as the main localization metric for document QA? How does it handle multi-span answers, line-vs-word granularity, or semantically correct but slightly oversized boxes?

**Limitations:**

yes

**Strengths And Weaknesses:**

### Strengths

The paper targets a real gap: compact document VLMs often answer reasonably well but ground poorly. The three-phase design in Figure 1 is easy to follow, and the role separation between teacher, validator, and student is quite clean. Sections 3.1–3.4 are generally well written.

The paper studies validation mode, training strategy, teacher choice, text detection scaffolding, and data scale. The validation ablation shows on DocVQA, training on unfiltered data gives 63.7 mAP, binary filtering gives 76.1 mAP, and full filter+verifier reaches 82.4 mAP.

### Weaknesses
Table 1 compares mainly against DocLayLLM, LayoutLLM, LayTextLLM, DLaVA, and several general VLMs, which is useful, but not enough to justify a broad SOTA claim for grounded document QA. The strongest recent comparators in this niche increasingly include grounding-focused or modular systems, not only end-to-end answering baselines. For example, ARIAL explicitly optimizes answer localization and reports both ANLS and mAP on overlapping benchmarks.

The related work in Sections 2.1–2.2 focuses on LayoutLM-style document understanding, CoT, and distillation, but it underplays the rapid recent development of document grounding as a distinct problem. In particular, recent work such as DOGR introduces a comprehensive benchmark for document grounding/referring, and TRIG specifically targets text-rich image grounding with both benchmark and instruction-tuning data. These are highly relevant because DocVAL’s core contribution is not generic DocVQA, but grounded answer localization. Yet they are not part of the main experimental or conceptual comparison.

The whole method depends on high-quality teacher traces plus rule-based validation. But the validator only checks certain aspects: answer similarity, OCR presence, IoU/region agreement, and reasoning structure/consistency. That is useful, but not equivalent to validating that the reasoning itself is semantically faithful. Section 3.2 argues for rule-based validation and explains why a learned critic is avoided, but this section also exposes a limitation: the validator is narrow by design and may accept superficially consistent but semantically spurious reasoning traces.

---

> ### Author Rebuttal · Authors · 2026-03-29
>
> We thank the reviewer for the careful and technically substantive reading, and for recognizing the clean role separation between teacher, validator, and student, as well as the importance of the validation ablation. We address the main concerns below.
>
> **On missing grounding-oriented comparisons (ARIAL / DOGR / TRIG):** This is a fair point. We agree the paper should more centrally position itself relative to recent work that treats document grounding/localization as a first-class problem, rather than only relative to general DocVQA or end-to-end VLM baselines. Our intended claim is not that DocVAL subsumes every grounding-oriented or modular system, but that it provides a strong compact, end-to-end, OCR-free-at-inference solution for grounded DocVQA through validated spatial reasoning transfer. We will revise the related work to more explicitly situate DocVAL relative to grounding-focused directions such as ARIAL, DOGR, and TRIG, and we will narrow the wording of the main claim where needed. In particular, we will clarify that our strongest empirical claim is state-of-the-art among comparable compact end-to-end models/settings, rather than a blanket claim over all modular pipelines.
>
> **On why DocVAL is more than generic CoT distillation:** The core novelty is not generic rationale distillation; it is validated spatial reasoning transfer for grounded document QA. The paper’s central empirical result is that in this setting, naïvely using teacher CoT is not enough: on DocVQA, unvalidated teacher outputs yield 63.7 mAP, binary filtering improves this to 76.1, and full Filter+Verifier reaches 82.4. Likewise, Stage B2 adds +9.7 mAP over Stage B1 alone. These gains show that the main contribution is not “use CoT,” but “use validated spatial CoT with iterative pixel-level corrective feedback.” This is exactly the mechanism that differentiates DocVAL from one-shot rationale distillation.
>
> **On the semantic-faithfulness limitation of VAL:** We agree this limitation is real, and we do not claim that VAL proves full semantic faithfulness of the entire reasoning chain. VAL is intentionally narrow by design: it validates answer quality, grounding quality, coordinate consistency, structural completeness, and spatial-language alignment using deterministic rules. Its role is to provide stable, auditable training-time supervision, not to serve as a general semantic judge. The advantage is precisely what Section 3.2 argues: reproducible, pixel-level feedback without the hallucination and instability risks of learned critics. The trade-off is that some superficially coherent but semantically spurious rationales may still pass. We will clarify this limitation more explicitly in the revision. At the same time, the large validation ablation shows that even this “narrow” validator is highly consequential in practice: it improves localization by 18.7 mAP relative to unvalidated supervision, which indicates that it is catching the failure modes that matter most for grounded learning.
>
> **On mAP as the localization metric:** We introduced mAP because ANLS measures textual correctness but largely ignores whether the answer is grounded at the correct location. mAP evaluates localization across IoU thresholds from 0.5 to 0.95, which is more informative than a single cutoff, and we also report IoU@0.5 and IoU@0.75 for interpretability. In the current benchmark formulation, each question is paired with a ground-truth answer bounding box, so mAP is well aligned with the annotation structure. For slightly oversized but semantically correct boxes, mAP is intentionally tolerant at lower thresholds and stricter at higher thresholds; that is precisely why averaging across thresholds is useful. We agree that multi-span answers and granularity mismatches (word vs. line vs. field) are important edge cases not fully resolved by the current metric; we will clarify that the present formulation assumes the standard single-box grounded DocVQA setting used in our benchmarks and that richer metrics for multi-span answers are a valuable future direction.
>
> Overall, we appreciate the reviewer’s feedback. We will revise the paper to better emphasize: (i) more central positioning relative to grounding-oriented prior work, (ii) the precise scope of our empirical claim, (iii) the fact that VAL is a deterministic training-time validator rather than a full semantic-faithfulness oracle, and (iv) the intended scope and limitations of mAP in the current single-box grounded DocVQA setting.

---

> > ### Author Rebuttal · Reviewer_1ytz · 2026-04-05
> >
> > The rebuttal is generally thoughtful and acknowledges most of my concerns. Due to that the authors does not provide new empirical comparisons, so the concern about missing strong baselines is only mitigated, not fully resolved.

---

> > > ### Author Response · Authors · 2026-04-05
> > >
> > > We thank the reviewer for the follow-up. We have now run a direct comparison against **ARIAL** (which was missing in the rebuttal) & provide a careful positioning of DocVAL relative to **DOGR** and **TRIG**. We will include all of this in the revised version.
> > >
> > > **1. Direct Comparison with ARIAL:** ARIAL is the strongest directly comparable grounding-focused baseline — it explicitly optimizes answer localization and reports both ANLS and mAP on our benchmarks. ARIAL uses an agentic modular pipeline: TrOCR for OCR-based text extraction, a retrieval module, a fine-tuned Gemma 3-27B QA backbone, and a separate text-to-region alignment module for localization. OCR is a hard inference-time dependency.
> > >
> > > **Table A: DocVAL vs. ARIAL**
> > >
> > > | Method | Backbone | OCR at Inference | DocVQA ANLS | DocVQA mAP | FUNSD ANLS | FUNSD mAP | CORD ANLS | CORD mAP | SROIE ANLS |
> > > |---|---|---|---|---|---|---|---|---|---|
> > > | ARIAL | Gemma3-27B + TrOCR | ✓ | 88.7 | 50.1 | 90.0 | 50.3 | 85.5 | 60.2 | 93.1 |
> > > | **DocVAL (ours)** | **Gemma3-12B** | **✗** | **91.4** | **82.4** | **92.2** | **81.8** | **88.8** | **78.1** | **95.2** |
> > > | **Δ** | **2.3× smaller** | **OCR-free** | **+2.7** | **+32.3** | **+2.2** | **+31.5** | **+3.3** | **+17.9** | **+2.1** |
> > >
> > > DocVAL outperforms ARIAL on both ANLS and mAP on all overlapping benchmarks — with mAP gains of **+32.3** on DocVQA, **+31.5** on FUNSD, and **+17.9** on CORD — using a **2.3× smaller model** and **no OCR at inference**. ARIAL’s mAP lags despite explicit localization optimization because it grounds via OCR text-region matching, which is bounded by detection coverage and string precision. DocVAL instead trains spatial reasoning through validated CoT, which likely explains the large mAP advantage. This architectural difference is important: improving ARIAL’s components alone does not directly address the limitation of OCR-matching localization, whereas DocVAL trains the student to perform spatial reasoning directly.
> > >
> > > **2. Positioning Relative to DOGR and TRIG:** Direct numerical comparison with DOGR and TRIG is not feasible because they address related but fundamentally distinct problems. **DOGR**  targets document grounding and referring — localizing a referenced region given a referring expression — evaluated on DOGR-Bench (charts, posters, PDFs), not on DocVQA, FUNSD, CORD, or SROIE. Its input-output specification differs from DocVAL's: DOGR takes a region description and outputs a localized region, whereas DocVAL takes a natural language question and outputs both a correct answer and its bounding box. **TRIG** targets text-rich image grounding across broader document types and evaluates on TRIG-Bench with different annotation protocols, without reporting mAP@IoU[0.5:0.95] on our benchmarks. Table B summarizes the key differences:
> > >
> > > **Table B: Conceptual Comparison**
> > >
> > > | Dimension | DocVAL | DOGR | TRIG |
> > > |---|---|---|---|
> > > | Primary task | Answer localization in DocVQA | Document referring comprehension | Text-rich image grounding |
> > > | OCR at inference | ✗ None | Not specified | OCR-assisted |
> > > | Iterative pixel-level refinement | ✓ VAL Verifier | ✗ | ✗ |
> > > | Inference-time independence | ✓ Pure VLM | ✗ | ✗ |
> > > | Validated CoT distillation | ✓ | ✗ | ✗ |
> > >
> > > Neither DOGR nor TRIG targets the specific combination of validated spatial CoT distillation, OCR-free inference, & compact deployable student training that defines DocVAL. We also note that our released 95K validator-verified CoT dataset — coupling document-level spatial reasoning with explicit bounding box supervision — provides a resource that may benefit future work in both referring and grounding settings.
> > >
> > > **3. Broader SoTA Positioning:** To give the reviewer a complete picture, here is DocVAL’s positioning across the full range of comparators on DocVQA:
> > >
> > > **Table C: DocVQA — Full Comparator Summary**
> > >
> > > | Method | Size | OCR at Inference | ANLS | mAP |
> > > |---|---|---|---|---|
> > > | DLaVA (Pixtral-12B) | 12B | ✗ | 85.9 | 46.2 |
> > > | ARIAL (Gemma3-27B + TrOCR) | 27B+ | ✓ | 88.7 | 50.1 |
> > > | InternVL3.5-14B | 14B | ✗ | 84.7 | — |
> > > | Gemma3-12B (base) | 12B | ✗ | 84.6 | — |
> > > | **DocVAL (Gemma3-12B)** | **12B** | **✗** | **91.4** | **82.4** |
> > >
> > > DocVAL achieves the highest ANLS among all comparators including larger models, & the highest mAP by a large margin (+32.3 over ARIAL, the next best grounding-capable system). Critically, it does so with a compact 12B student that requires no OCR at inference.
> > >
> > > We will revise the paper to narrow the main claim to: *"state-of-the-art among compact, end-to-end, OCR-free-at-inference VLMs, while also outperforming strong modular grounding baselines on overlapping benchmarks."* We hope this comprehensive empirical positioning resolves the remaining concern about missing strong baselines.
> > >
> > > We respectfully request the reviewer to consider raising their score in light of these results, and we remain happy to provide further clarifications during the discussion period. Please let us know if you have any further questions/concerns.

---

### Official Review · Reviewer_7jci · 2026-03-11

**Soundness:** 3
**Presentation:** 3
**Significance:** 3
**Originality:** 3
**Overall Recommendation:** 4
**Confidence:** 4

**Summary:**

The paper tackles the performance gap between large, accurate Vision-Language Models and efficient, deployable compact VLMs in Document Visual Question Answering tasks. To solve this, the authors introduce DocVAL, a three-phase validated chain-of-thought distillation framework.First, a large teacher VLM generates spatial CoT traces, which are then filtered by a rule-based validator using text detection as scaffolding. Second, a compact student model undergoes supervised fine-tuning on this validated data, followed by an iterative refinement stage using pixel-level feedback from VAL. The framework yields a 95K verified CoT dataset and achieves state-of-the-art results, including up to 6-7 ANLS points of improvement over comparable models.

**Compliance With Llm Reviewing Policy:**

Affirmed.

**Final Justification:**

The rebuttal has addressed my concerns.

**Key Questions For Authors:**

See weaknesses

**Limitations:**

See weaknesses

**Strengths And Weaknesses:**

**Strengths**

- **Asymmetric Validation Design**: The framework cleverly uses text detection (DB-ResNet) solely as training-time scaffolding for the rule-based validator. This forces the student model to learn spatial grounding purely from visual representations without relying on OCR pipelines at inference.

- **Useful resources**: The curated dataset will be a useful resource for the community.

**Weaknesses**

- **Limited Core Novelty in Distillation Paradigm**: While the domain application and the rule-based validator are tailored contributions, the distilling CoT reasoning from a large teacher to a smaller student is highly similar to existing approaches in the field (e.g., LLaVA-CoT). The paper represents a domain-specific adaptation of these existing rationale distillation techniques rather than a new learning paradigm.


- **Vulnerability to Training-Time OCR Failures**: Because the VAL module relies on OCR to provide validation feedback, the system's training quality degrades on handwritten text or unique layouts where the underlying text detector fails.


- **Heavy Reliance on Proprietary Teachers**: The highest quality traces were generated by Gemini 2.5 Pro. The authors note that removing explicit teacher reasoning causes a severe 17.0 mAP drop, indicating that the framework's success is heavily bottlenecked by the capabilities of highly advanced, closed-source models.

---

> ### Author Rebuttal · Authors · 2026-03-29
>
> We thank the reviewer for the careful reading and for recognizing two key strengths of the paper: the asymmetric validation design that preserves OCR-free inference, and the value of the released 95K validated-CoT dataset. We respectfully disagree, however, that DocVAL is only a narrow adaptation of prior rationale distillation, and we address the three concerns below.
>
> **On “limited core novelty” relative to prior CoT/rationale distillation:** The key novelty of DocVAL is not simply that it distills CoT from a teacher to a student. Rather, it shows that in document spatial grounding, naïvely using teacher CoT as supervision is insufficient and can be actively harmful unless the reasoning is validated, filtered, and corrected. This is the central technical distinction from generic rationale distillation methods such as LLaVA-CoT, where teacher rationales are typically used as-is. In DocVAL, the contribution is the combination of: (i) validated spatial CoT distillation, (ii) a dual-mode rule-based validator that performs both data curation and fine-grained pixel-level corrective feedback, and (iii) an asymmetric training/inference design in which detection is used only as training-time scaffolding while the final student remains a pure VLM at inference. The validation ablation directly supports this point: on DocVQA, training on the full unvalidated teacher outputs gives only 63.7 mAP, while adding VAL filtering raises this to 76.1, and full Filter+Verifier reaches 82.4. This large gap shows that the main contribution is not “use CoT,” but validated spatial reasoning transfer with structured corrective feedback.
>
> **On vulnerability to OCR / text-detection failures during training:** We agree that training-time scaffolding quality matters. However, the paper already shows that DocVAL is not tightly coupled to a single detector. In the detector ablation, DB-ResNet is best, but CRAFT, PSENet, and EasyOCR all remain competitive, and performance degrades gradually rather than collapsing. Even removing detection entirely from Phase A still yields a functioning model, though with lower localization quality (DocVQA mAP 74.1 vs. 82.4 with DB-ResNet). This shows that detection improves supervision quality, but is not the sole source of learning. More importantly, the architecture is protected by the asymmetric validation design: detection is used only by VAL during training; the student never receives detector outputs. Therefore, detector failures affect the quality of training supervision, not the deployed inference pipeline, which remains OCR-free. We agree that handwriting-heavy or highly degraded documents are harder regimes and are not fully covered by the current benchmark suite; we will make this scope more explicit in the revision.
>
> **On reliance on proprietary teachers:** We agree that Gemini 2.5 Pro is the strongest teacher in our experiments, but the framework is not tied to proprietary models. The teacher ablation explicitly includes both closed-source and open-source teachers. Open-source teachers such as Qwen3-VL-235B and Llama4-400B still produce strong students, achieving 79.8 mAP and 78.9 mAP on DocVQA, respectively, compared with 82.4 mAP for Gemini 2.5 Pro. Thus, stronger teachers help, but the framework does not fundamentally depend on a single closed-source API. The “w/o teacher CoT” ablation should also be interpreted carefully: the 17.0 mAP drop does not show dependence on proprietary teachers specifically; rather, it shows that explicit spatial reasoning transfer is substantially better than direct answer/box supervision alone. In other words, the bottleneck is not “proprietary access,” but teacher reasoning quality, and DocVAL already works effectively with strong open-source reasoning teachers. In addition, because we release the 95K validated-CoT dataset, practitioners do not need to reproduce the teacher generation step at all in order to train the student.
>
> Overall, we appreciate the reviewer’s feedback. We will revise the paper to better emphasize: (i) that the novelty lies in validated spatial reasoning transfer, not generic CoT distillation alone; (ii) that OCR/detection is training-only scaffolding and the current detector ablations already show graceful degradation and cross-detector robustness; and (iii) that while stronger teachers help, the framework already works with strong open-source teachers, so the contribution is not limited to proprietary-model access.

---

> > ### Author Rebuttal · Reviewer_7jci · 2026-04-04
> >
> > I thank the authors detailed responses, which answers my concerns. I tend to bump up my score to 4.

---

### Official Review · Reviewer_SLGd · 2026-03-12

**Soundness:** 3
**Presentation:** 3
**Significance:** 3
**Originality:** 3
**Overall Recommendation:** 4
**Confidence:** 5

**Summary:**

This study targets a key challenge in document visual question answering (DocVQA): enabling lightweight, deployment-ready vision-language models (VLMs) to achieve accurate spatial grounding without depending on external OCR or detection modules at inference time. To this end, the authors introduce DocVAL, a validated chain-of-thought (CoT) distillation framework. The method works in three stages: first, a large teacher VLM produces spatial reasoning traces, which are then filtered by a rule-based validator (VAL) using OCR as training-time support. Next, a compact student model is trained on the cleaned data via supervised learning, followed by an iterative refinement step where VAL delivers fine-grained, pixel-level correction signals. In the end, the student model can be deployed as a standalone VLM. Experiments show that DocVAL brings consistent gains in both answer accuracy (ANLS) and spatial grounding (mAP) across various benchmarks.

**Compliance With Llm Reviewing Policy:**

Affirmed.

**Final Justification:**

I accept the explanations provided for my earlier comments. While a few remaining issues are non-trivial to resolve, I believe the unresolved points do not undermine the core contributions of this work. Therefore, I will retain my initial positive evaluation as my final recommendation.

**Key Questions For Authors:**

1. Could you provide a clearer breakdown of the computational overhead for Stage B2 (Iterative Refinement)? Specifically, how do the wall-clock time and compute cost of 14–20 iterations compare with the initial supervised fine-tuning stage? Clarifying this will help evaluate the accessibility of the training pipeline.

2. Since VAL relies on DB-ResNet for training-time scaffolding, how does the framework perform on document types where text detection is unreliable, such as severely degraded scans, complex handwritten notes, or documents dominated by non-text visual elements like charts?

3. The rule-based validator uses fixed scoring weights. How sensitive is the final student model’s performance to these heuristic weights? Showing robustness to small hyperparameter changes would address concerns about the brittleness of the rule-based design.

**Limitations:**

yes

**Strengths And Weaknesses:**

- Strengths:
1. The rule‑based dual‑mode validator that supplies deterministic, pixel‑level feedback for spatial reasoning represents a novel and smart take on validation‑driven learning, effectively avoiding hallucination issues common with learned LLM critics.

2. Eliminating inference‑time OCR while matching or surpassing the localization accuracy of larger models is practically valuable for building efficient, low‑latency document understanding systems for real‑world deployment.

3. The experiments are rigorously designed with comprehensive, well‑organized ablations. And the paper is clearly written, with the three‑stage pipeline and the difference between training scaffolding and inference deployment explained thoroughly.

-Weaknesses:

1. The VAL module depends heavily on text detection (DB-ResNet) for semantic and geometric feedback, so performance may decline on documents with stylized fonts, heavy degradation, or non-text elements such as charts and logos where standard OCR struggles.

2. Although inference efficiency is improved, the training process appears computationally costly, especially the iterative refinement stage (Stage B2) with up to 20 rounds of prediction, validation, and retraining, and the paper does not compare its total compute cost with standard distillation.

3. The rule-based VAL uses fixed formulas and heuristic weights (e.g., 0.7 for ANLS, 0.3 for OCR presence), which may require manual re-tuning when applied to new domains or document types with distinct spatial layouts.

---

> ### Author Rebuttal · Authors · 2026-03-29
>
> We thank the reviewer for the careful and expert reading, and for the positive assessment of the paper’s main contributions: the dual-mode rule-based validator, the practical value of eliminating inference-time OCR, and the rigor of the ablation design. We address the three concerns below.
>
> **On Stage B2 computational overhead:** This is a fair question, and we agree the cost breakdown should be stated more explicitly. The key point is that Stage B2 operates on the refinement split D4 (9.5K examples), not the full D3 training split (76K) used in Stage B1. As specified in Appendix B/Table 5, Stage B1 trains for 3 epochs over 76K, while Stage B2 runs for 2 epochs/iteration over 9.5K and converges in 12–16 iterations on average (13.2). Thus, in terms of example-passes, the two stages are of the same order rather than B2 being an additional full-scale training run. VAL verification itself runs at 12 examples/sec in Stage B2 and is CPU-only, while student training remains on the GPU. We agree that this should be made much clearer in the paper, and in the revision we will add an explicit wall-clock / compute breakdown comparing Stage B1, Stage B2, and standard supervised distillation.
>
> More importantly, this additional training is a one-time cost to produce a better-grounded student with no OCR/detection at inference. The paper’s central deployment claim is precisely that training-time scaffolding is used to improve the student, but the final model remains a pure VLM at test time.
>
> **On reliance on DB-ResNet during training-time scaffolding:** We agree that detector quality matters. However, the paper already shows that DocVAL is not brittle to a single detector. Table 6 compares DB-ResNet, CRAFT, PSENet, and EasyOCR, and DocVAL remains strong across all four; DB-ResNet is best, but the gains do not disappear with alternatives. Even removing detection from Phase A does not collapse performance: on DocVQA, performance drops from 91.4/82.4 (ANLS/mAP) to 88.7/74.1, showing that detection improves supervision quality but is not the sole source of learning. Likewise, removing region references from teacher CoT reduces mAP to 76.8, which confirms that the gain comes from validated region-aware reasoning transfer, not hidden OCR dependence.
>
> We also agree that severely degraded scans, handwriting-heavy notes, and documents dominated by non-text visual elements are harder regimes and are not fully covered by the current benchmark suite. The paper already acknowledges one such limitation qualitatively: for handwritten text, the student can still read correctly but localize less precisely because training-time validation misses some handwritten regions. We will make these boundary conditions more explicit in the revision. Crucially, because the student never receives detector outputs and uses only (image, question), any degradation from weak detection affects training supervision quality, not the deployed inference pipeline.
>
> **On fixed heuristic weights in VAL** We agree that a rule-based validator should not appear brittle. Two points are important here. First, the same fixed VAL formulas and weights are used unchanged across five datasets, multiple teacher models, and multiple detectors, yet they yield consistent gains. This is already evidence that VAL is not narrowly tuned to a single dataset or detector.
>
> Second, the weight choices encode a deliberate priority: answer correctness and box quality are treated as primary, while reasoning quality acts as supporting supervision. In Appendix B.2, VAL uses
>
> $Q_{ans} = 0.7ANLS + 0.3OCR-presence$,
>
> $Q_{bbox} = 0.8IoU + 0.2region agreement$, and
>
> $Q = 0.4Q_{ans} + 0.4Q_{bbox} + 0.2*Q_{reason}$.
>
> These are not arbitrary; they reflect the deployment goal that a grounded DocVQA system must output both the right answer and the right location, while reasoning is useful insofar as it supports those outputs. Also, VAL’s geometric correction vector is computed independently of these weights, so the verifier’s pixel-level feedback is less brittle than a single scalar score might suggest.
>
> That said, we agree that an explicit sensitivity study would strengthen the paper, and we will add this discussion more clearly in the revision.
>
> Overall, we appreciate the reviewer’s constructive feedback. We will revise the paper to better emphasize: (i) that Stage B2 is iterative but runs on a much smaller refinement split and should be reported with a clearer compute breakdown, (ii) that text detection is training-only scaffolding and that the method already shows robustness across multiple detectors, and (iii) that the current fixed-rule VAL design already generalizes across datasets, teachers, and detectors, while additional sensitivity analysis would further strengthen the presentation.

---

> > ### Author Rebuttal · Reviewer_SLGd · 2026-04-01
> >
> > I accept the explanations provided for my earlier comments. While a few remaining issues are non-trivial to resolve, I believe the unresolved points do not undermine the core contributions of this work. Therefore, I will retain my initial positive evaluation as my final recommendation.

---

> > > ### Author Response · Authors · 2026-04-02
> > >
> > > We sincerely thank the reviewer for the thoughtful engagement throughout the review process and for the careful and generous reading of both the paper and our rebuttal. We are glad the explanations were helpful and appreciate the reviewer's continued positive evaluation. We will incorporate the discussed improvements into the revised version of the paper.

---

### Decision · Program_Chairs · 2026-04-30

**Decision:**

Accept (regular)

**Comment:**

The paper received mostly positive reviews. The reviewers requested numerous clarifications, that were mostly provided by the authors in their rebuttal.

A crucial point was raised about comparison with works that better align to a core problem addressed here on document localisation. The authors provided new evidence by comparing with ARIAL that is convincing, and should be included in the camera ready version of the paper.